# Rotational dynamics in motor cortex are consistent with a feedback controller

**Hari Teja Kalidindi[1†], Kevin P Cross[2]\*[†], Timothy P Lillicrap[3], Mohsen Omrani[2], Egidio Falotico[1], Philip N Sabes[4], Stephen H Scott[2]**

[1]The BioRobotics Institute, Scuola Superiore Sant'Anna, Pisa, Italy; [2]Centre for Neuroscience Studies, Queen's University, Kingston, Canada; [3]Centre for Computation, Mathematics and Physics, University College London, London, United Kingdom; [4]Department of Physiology, University of California, San Francisco, San Francisco, United States

**Abstract** Recent studies have identified rotational dynamics in motor cortex (MC), which many assume arise from intrinsic connections in MC. However, behavioral and neurophysiological studies suggest that MC behaves like a feedback controller where continuous sensory feedback and interactions with other brain areas contribute substantially to MC processing. We investigated these apparently conflicting theories by building recurrent neural networks that controlled a model arm and received sensory feedback from the limb. Networks were trained to counteract perturbations to the limb and to reach toward spatial targets. Network activities and sensory feedback signals to the network exhibited rotational structure even when the recurrent connections were removed. Furthermore, neural recordings in monkeys performing similar tasks also exhibited rotational structure not only in MC but also in somatosensory cortex. Our results argue that rotational structure may also reflect dynamics throughout the voluntary motor system involved in online control of motor actions.

**\*For correspondence:**
13kc18@queensu.ca

[†]These authors contributed equally to this work

## Editor's evaluation

Motor cortical population activity during reaching exhibits rotational dynamics thought to arise from recurrent connections in cortical circuits. This innovative paper performs an important 'experiment' not currently possible in real biological networks: examine activity and task function before and after deletion of recurrent connections. Surprisingly, trained networks produced rotational dynamics even without internal recurrence, raising the possibility that sensory feedback is a key determinant of motor cortical dynamics. More broadly, this paper leverages the experimental tractability of artificial neural networks to test what conditions and architectures are necessary to produce brain-like signals.

## Introduction

Motor cortex (MC) plays an important role in our ability to make goal-directed motor actions such as to reach and grasp objects of interest in the environment. A key approach to explore MC's contribution to movement has been to record the patterns of neural activity during tasks such as reaching. In the last part of the 20th century, research emphasized the representation of movement parameters by cortical networks (*Andersen and Buneo, 2002*; *Fetz, 1992*; *Kalaska and Crammond, 1992*; *Scott, 2008*). This approach assumed that the activity of individual neurons or at the population level could be directly related to explicit features of motor action such as movement speed or muscle activity patterns.

However, there has been a recent transition toward interpreting neural processing using dynamical systems techniques (*Machens et al., 2010*; *Michaels et al., 2016*; *Pandarinath et al., 2018b*; *Pandarinath et al., 2018a*; *Remington et al., 2018*; *Russo et al., 2018*; *Sauerbrei et al., 2020*; *Shenoy et al., 2013*; *Suresh et al., 2020*; *Vyas et al., 2020*). *Churchland et al., 2012* recorded from MC while monkeys performed goal-directed reaches and fit the population activity to a dynamical system where future activity was predicted based solely on the past population activity in MC. They found this relationship could account for a significant amount of the neural activity and revealed rotational dynamics that could provide a basis set for generating the necessary patterns of muscle activity.

The view of MC as a pattern generator during reaching was bolstered by recurrent neural network (RNN) models (*Hennequin et al., 2014*; *Kao et al., 2021*; *Logiaco et al., 2021*; *Michaels et al., 2016*; *Sussillo et al., 2015*). RNNs trained to generate patterns of muscle activity while constrained to generate simple dynamics also displayed rotational dynamics that resembled MC activity (*Sussillo et al., 2015*). Importantly, these networks only received external inputs that were stationary with the exception of a non-selective GO cue to initiate the pattern generation. Thus, activity was generated solely by the intrinsic connections between neurons, and online feedback about the generated muscle patterns was not necessary after training (*Sussillo et al., 2015*). Collectively, the findings that MC dynamics are well described by a deterministic dynamical system and that RNNs with dynamics dominated by intrinsic connections can approximate MC dynamics have led to a common interpretation that these dynamics reflect a pattern generator for muscle activity and that this real-time process is done largely autonomously from other brain structures and sensory feedback (*Bizzi and Ajemian, 2020*; *Collinger et al., 2018*; *Diedrichsen and Kornysheva, 2015*; *Kalaska, 2019*; *Khanna et al., 2021*; *Pandarinath et al., 2018a*; *Perich et al., 2020*; *Rouse et al., 2018*; *Suminski et al., 2015*; *Suresh et al., 2020*; *Vyas et al., 2020*).

Another class of dynamical systems is also commonly used in motor control to interpret the behavioral aspects of motor actions. Specifically, a growing body of literature has highlighted how optimal feedback control (OFC) can capture how we move and interact in the world (*Franklin and Wolpert, 2011*; *Scott, 2004*; *Scott, 2016*; *Shadmehr and Krakauer, 2008*; *Todorov and Jordan, 2002*). OFC highlights the importance of feedback processes, both external sensory feedback (e.g., proprioception and vision) as well as internal feedback from efference copies, for generating motor commands for movement. A large number of studies inspired by OFC highlight how humans are capable of generating fast, goal-directed motor corrections (*Cluff and Scott, 2015*; *Cross et al., 2019*; *Diedrichsen, 2007*; *Dimitriou et al., 2012*; *Kurtzer et al., 2008*; *Nashed et al., 2014*; *Scott, 2016*) even for very small disturbances (*Crevecoeur et al., 2012*) and OFC can capture features of unperturbed movements (*Knill et al., 2011*; *Lillicrap and Scott, 2013*; *Liu and Todorov, 2007*; *Nashed et al., 2012*; *Todorov and Jordan, 2002*; *Trommershäuser et al., 2005*). Further studies highlight how feedback responses to a mechanical disturbance are distributed throughout somatosensory, parietal, frontal, and cerebellar motor circuits in ~20 ms and display goal-directed responses in as little as 60 ms (*Chapman et al., 1984*; *Conrad et al., 1975*; *Cross et al., 2021*; *Evarts and Tanji, 1976*; *Herter et al., 2009*; *Lemon, 1979*; *Omrani et al., 2016*; *Phillips et al., 1971*; *Pruszynski et al., 2011*; *Pruszynski et al., 2014*; *Strick, 1983*; *Wolpaw, 1980*). Finally, a recent study demonstrates that inputs from motor thalamus to MC are essential for the execution of motor actions (*Sauerbrei et al., 2020*). This interpretation of motor control emphasizes that the objective of the motor system is to attain the behavioral goal and this requires feedback processed by a distributed network. Further, MC is generally viewed as part of the control policy that uses information on the system state to generate muscle activity to attain the behavioral goal.

These two views of MC, one as an autonomous dynamical system and the other as a flexible feedback controller, appear to conflict on how to interpret the role of MC and its interactions with the rest of the neural circuit and sensory feedback involved in goal-directed motor actions. This apparent conflict seems to hinge on the observation that the rotational dynamics observed in MC can be generated through purely intrinsic recurrent connections. However, it is unclear if a network with external feedback would also exhibit similar rotational dynamics and whether these dynamics are exclusively in MC or also in other brain regions such as somatosensory cortex. We investigated this question by first developing a multi-layer RNN that controlled and received sensory feedback from a two-segment limb. The network was trained to counter disturbances to the limb and perform reaching movements. After training, rotational dynamics were observed in the network activities as well as in

sensory feedback from the limb, but not in muscle activity. Critically, rotational dynamics were generated in networks trained with and without intrinsic recurrent connections. Monkeys trained in similar tasks exhibited rotational dynamics in MC and in somatosensory and parietal cortices including during reaching where sensory feedback is not required a priori. Taken together, these results illustrate rotational dynamics can be observed across frontoparietal networks and can be generated by intrinsic dynamics in MC and/or through dynamics of the entire motor system.

## Results

### RNN exhibit rotational dynamics in the activities and sensory feedback signals during posture task

One interpretation of rotational dynamics is that it provides a signature of an autonomous dynamical system. In contrast, rotational dynamics appear to be absent in systems dominated by external inputs, such as muscle activity driven by neural inputs (**Churchland et al., 2012**), or MC activity during grasping driven by sensory inputs (**Suresh et al., 2020**). The absence of rotational dynamics in input-driven systems can occur since these networks need not adhere to any dynamical roles to generate activity patterns. Instead, all necessary dynamics are generated by the inputs to the network and these activity patterns can be largely arbitrary or unstructured. Here, we examined the dynamics of a network performing a posture perturbation task, where the network had to respond to sensory feedback about the periphery to generate an appropriate motor correction (**Cross et al., 2020**; **Heming et al., 2019**; **Omrani et al., 2014**; **Omrani et al., 2016**; **Pruszynski et al., 2014**). Sensory input plays an important role for correctly performing the task, and thus, the hypothesis is that rotational dynamics should be absent in the network.

We first explored this hypothesis using trained neural networks where we could specify exactly the architecture of the network and the nature of the sensory feedback inputs. We built an artificial neural network that controlled a two-link model of the upper limb (**Figure 1A and B**). Previous neural network models (**Hennequin et al., 2014**; **Michaels et al., 2016**; **Sussillo et al., 2015**) focused on network activities (r) that evolved according to $\dot{r}(t) = f(r(t), s^*)$ where $f[\cdot]$ is a nonlinear function and $s^*$ is a vector of static inputs about the GO cue and the current target. Here, we generated a model where network activities also incorporated delayed (Δ) continuous sensory feedback about the limb (s(t-Δ)), and thus, activities evolved according to $\dot{r}(t) = f(r(t), s^*, s(t - \Delta))$. The neural network contained an input layer that had intrinsic recurrent connections between neurons and received delayed (Δ = 50 ms) sensory feedback about the limb state (i.e., joint position, velocity, and muscle activities). This layer projected to an output layer that also had intrinsic recurrent connections between neurons. The output layer directly controlled the activities of six muscles (two sets of monoarticular muscles at the shoulder and elbow joints and two biarticular muscles) that generated limb movements. The network was trained to perform a posture perturbation task where the goal was to keep the limb within a specified target location, while countering randomly applied loads to the limb. We optimized the network by minimizing a cost function that penalizes the kinematic error between the target location and current limb position over the duration of the task. Note, the only input to the network was sensory input from the limb (i.e., no task-goal input; **Figure 1C**).

After optimization, we applied loads that displaced the limb by ~3 cm. The network generated corrections to the displacements with the hand reversing direction within 300–400 ms from the time of the applied load (**Figure 2A–C**). The network also maintained steady-state motor output for the remainder of the trial to counter the applied loads. **Figure 2D** shows the activity of the shoulder extensor muscle aligned to the load onset. An increase in muscle activity started 50 ms after the applied load, consistent with the delay in sensory feedback from the limb. Muscle activity peaked at ~200 ms after the applied load and stabilized to a steady state within ~750 ms. **Figure 2E and F** show the activity of two example neurons from the output layer of the network.

We examined the population dynamics of the output layer of the network by applying jPCA analysis (**Churchland et al., 2012**). Briefly, jPCA constructs a multi-dimensional matrix ($X(t)$, dimensions n × ct) which is composed of each unit's (n) activity patterns across time (t) and condition (c) (e.g., load combination or reach target). The matrix is reduced ($X_{Red}$) to a 6 × ct dimensional matrix using principal component analysis (PCA) to examine the dynamics exhibited by the dominant signals. This matrix is then fit to a constrained dynamical system $\dot{X}_{Red}(t) = M_{Skew}X_{Red}(t)$ where $\dot{X}_{Red}(t)$ is the

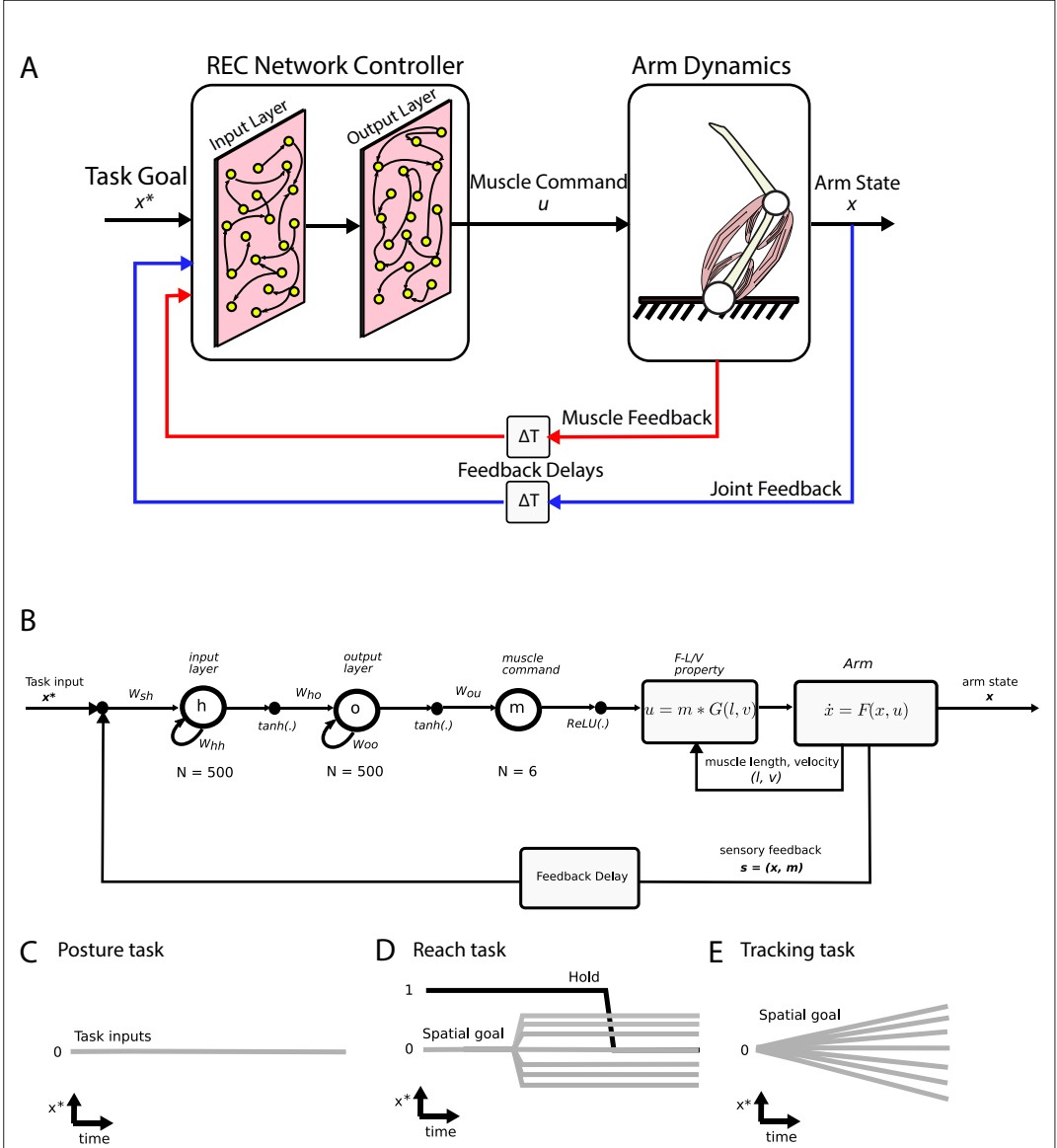

**Figure 1.** Simulation setup. (**A**) Schematic of the two-link model of the arm and the neural network. The arm had two joints mimicking the shoulder and elbow (arm dynamics: joints are white circles) and was actuated using six muscles (pink banded structures). Muscle activity was generated by the neural network (muscle command). The network was composed of two layers (input and output layers) with recurrent connections between units within each layer. The network received delayed (ΔT) sensory feedback from the limb in the form of joint angles and velocities (joint feedback, blue line), and muscle activities (muscle feedback, red line). Delays were set to 50 ms to match physiological delays. The network also received input about the desired location of the limb (task goal). (**B**) Computational graph for the same network depicted in (**A**). $W_{hh}$ and $W_{oo}$ are the recurrent connections for the input and output layers, respectively. For the NO-REC network, these connections were set to zero and remained at zero when optimizing. $W_{sh}$ are the connection weights between the inputs to the input layer and $W_{ho}$ are the connection weights between the input layer and the output layer. Tanh activation functions were used for the network layers and a rectified linear unit (ReLU) was used for the muscle layer. Muscle activity (m) was then converted to joint torques (u) while taking into account force-length (F-L) and force-velocity (F-V) properties of muscles. Joint torques were used to update the arm state (x) and sensory feedback (s) about the arm state and muscle activities was fed back into the input layer following a feedback delay. (**C–E**) Visual depictions of the task inputs to the network (x*) for each of the behaviours. There was no task input for the posture task (**C**). For the reach task (**D**), the task input reflected the spatial end position of the target as well as a GO cue (hold command, thick black line). For the tracking task (**E**), the task input reflected the spatial position of the moving target at each time point.

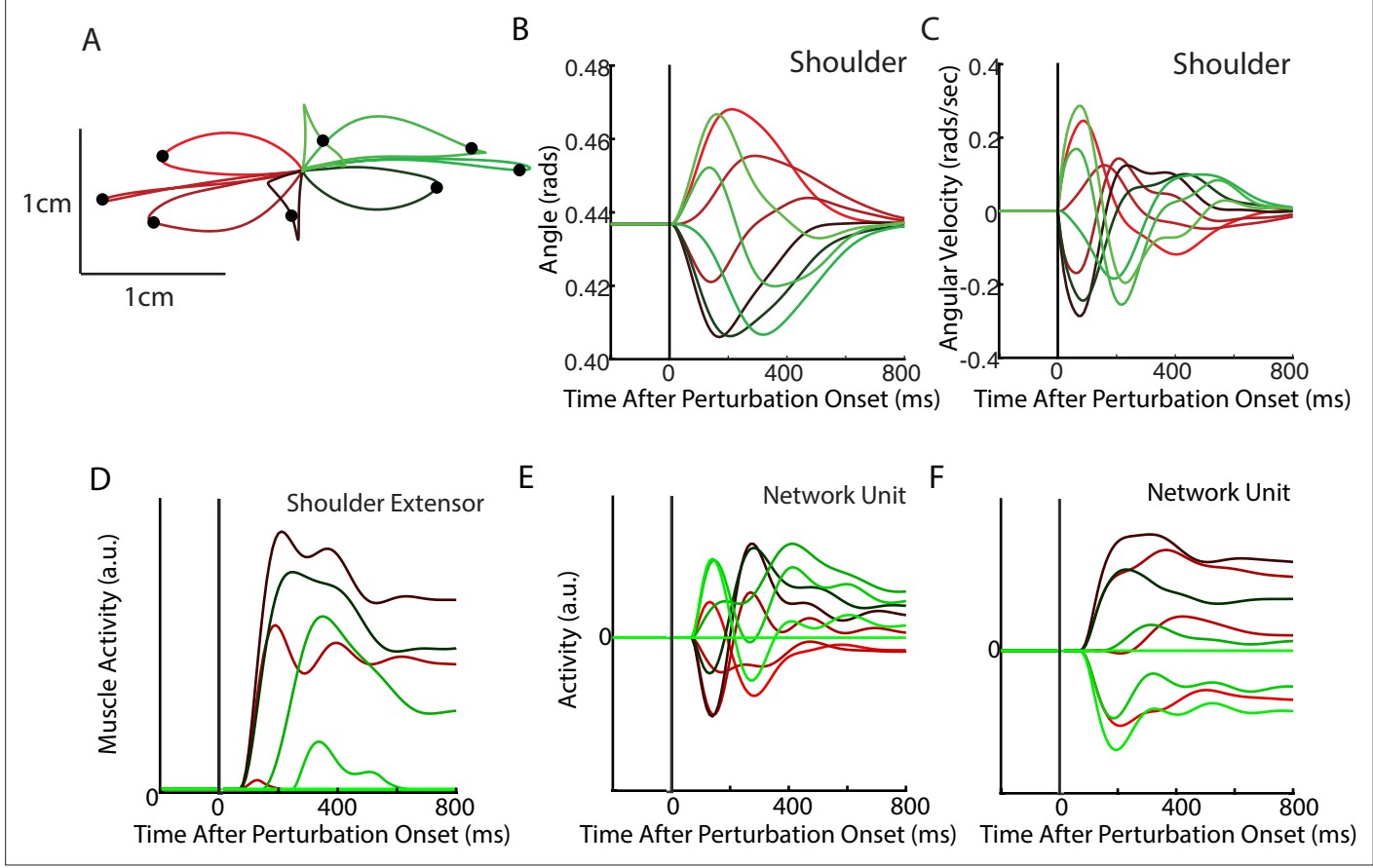

**Figure 2.** Posture perturbation task performed by neural network. (**A**) Hand paths when mechanical loads were applied to the model's arm. Due to the anisotropy in the biomechanics, the trajectories across the different loads are asymmetric. Black dots denote the hand's location 300 ms after the load onset. (**B, C**) Shoulder angle and angular velocity aligned to the load onset. (**D**) Activity of the shoulder extensor aligned to load onset. (**E, F**) The activities of two example units from the output layer of the network. The colors in (**A–F**) correspond to different directions of load.

temporal derivative of $X_{Red}(t)$, and $M_{Skew}$ is the weight matrix constrained to be skew-symmetric. The skew-symmetric constraint ensures that only rotational dynamics are fit to the population activity and $M_{Skew}$ can then be decomposed into a set of three jPC planes.

We found the top2 jPC planes exhibited clear rotational dynamics with rotation frequencies of 2.0 Hz and 0.7 Hz (*Figure 3A*, left and middle panels). Combined, these two planes captured  60% of the variance of the output-layer activities. In contrast, the third jPC plane exhibited a more expansion-like property (*Figure 3A*, right) and captured  38% of the variance. Note, jPC planes are orthogonal with respect to each other and are ranked by their eigenvalues from largest to smallest. These eigenvalues correspond to the rotational frequencies for each plane with larger eigenvalues corresponding to higher frequencies.

Examining the goodness of fit ($R^2$) to the constrained dynamical system provides a measure of how well the network activities are approximated by rotational dynamics. We compared our results to a null distribution that tested whether the rotational structure was an emergent property of the population activity or simply reflected known properties of single-neuron responses (i.e., broad tuning for loads, smooth time-varying activity patterns, and shared patterns of activity across neurons). We used tensor maximum entropy (TME; *Elsayed and Cunningham, 2017*) to generate surrogate data sets that were constrained to have the same covariances as the observed data and applied the same jPCA analysis to the data sets. We found the constrained dynamical system had an $R^2$ of 0.55 and was significantly greater than expected from the null distributions (*Figure 3B*, left; TME: median $R^2$=0.27, p=0.001). Further, when we did not constrain the weight matrix to be skew-symmetric (i.e., unconstrained dynamical system, $M_{Best}$), we found an increase in the $R^2$ to 0.83 that was also significant (*Figure 3B*,

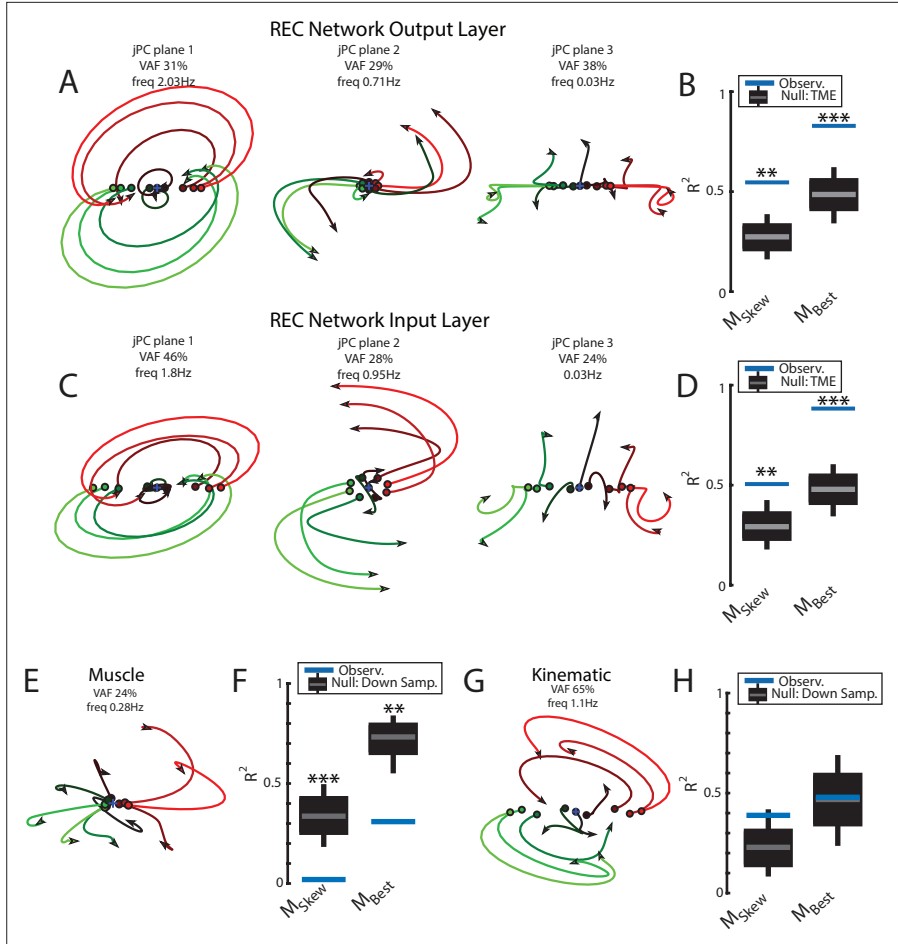

**Figure 3.** Population dynamics of the network during posture. (**A**) The top3 jPC planes from the activity in the output layer of the network. Dynamics were computed from 70 ms to 370 ms after the load onset. Different colors denote different load directions. (**B**) The goodness of fit (black horizontal line) of the network activity to the constrained ($M_{Skew}$ left) and unconstrained ($M_{Best}$ right) dynamical systems. Null distributions were computed using tensor maximum entropy (TME). Gray bars denote the median, the boxes denote the interquartile ranges, and the whiskers denote the 10th and 90th percentiles. (**C, D**) same as (**A, B**) except for the input layer of the network. (**E, F**) and (**G, H**) same as (**A, B**) except for the muscle activities and kinematic inputs into the network, respectively. Null distributions were computed from the down-sampled neural activity for (**F**) and (**H**). VAF, variance accounted for.

The online version of this article includes the following figure supplement(s) for figure 3:

**Figure supplement 1.** Decoding output layer trajectories using sensory input.

**Figure supplement 2.** Population dynamics in somatosensory cortex during posture task from *Chowdhury et al., 2020*.

right; median $R^2$=0.49, p<0.001). The ratio between the $R^2$ for the constrained and unconstrained fits was 0.66 indicating that the majority of the output layer's dynamics displayed rotational dynamics.

Next, we examined if rotational dynamics were present in the input layer of the network,which directly receives sensory feedback. Similar to the output layer, we observed rotational dynamics in the top-2 jPC planes with frequencies of 1.8 Hz and 0.95 Hz (*Figure 3C*). Combined, these two planes captured 74% of the variance of the inputlayer activity. The fit to a constrained dynamical system had an $R^2$ = 0.51 (*Figure 3D*, left) and was also significantly greater than the null distributions (median $R^2$=0.29, p<0.01). When fit with an unconstrained dynamical system, we also found an increase in the $R^2$ to 0.88 (*Figure 3D*, right) that was significantly greater than the null distributions (median $R^2$=0.48, p<0.001). Thus, rotational dynamics are present in the input layer that directly received sensory feedback as well as the output layer that formed the muscle signals.

Next, we explored if rotational dynamics were present in the motor outputs (i.e., muscle activities) and sensory inputs (i.e., muscle activities and joint kinematics) of the network. We applied jPCA analysis to the muscle activities and did not observe clear rotations in any of the jPC planes (*Figure 3E*). We found the muscle activities were poorly fit to the constrained (*Figure 3F*; $R^2$=0.02) and unconstrained dynamical systems ($R^2$=0.31). One explanation for this lower fit quality is that muscle activity has substantially fewer signals (6) than the network activities (500). We tested this by down-sampling neural units to match the number of muscles. Note, we did not compute a null distribution using TME as we found hypothesis testing using TME was unreliable when the number of signals was small (<30). We found the goodness of fits for muscle activities were significantly smaller than the down-sampled neural activities (*Figure 3F*, constrained p<0.001; unconstrained p=0.004) indicating that the down-sampled neural activity exhibited greater dynamical properties than muscle activity.

Next, we applied jPCA analysis to the kinematic signals (angle and angular velocity of the joints). We observed clear rotational dynamics in the top jPC plane (*Figure 3G*) with a rotational frequency of 1.1 Hz. We found the constrained and unconstrained dynamical systems had an $R^2$=0.39 and 0.48, respectively, which were comparable to the null distributions (*Figure 3H*; down-sampled neural population: constrained p=0.12 and unconstrained p=0.48).

These results indicate kinematic signals exhibit rotational dynamics comparable to neural activity. However, their rotational frequencies are lower than observed in the output layer activities. Here, we asked whether these higher frequencies could be explained by combining all available sensory feedback (i.e., muscle and kinematics). We fit a linear model that decoded the output layer's activity in each jPC plane using the sensory feedback signals composed of kinematic and muscle signals. We found the predicted activities were highly similar to the output layer activities ($R^2$=0.99) with virtually

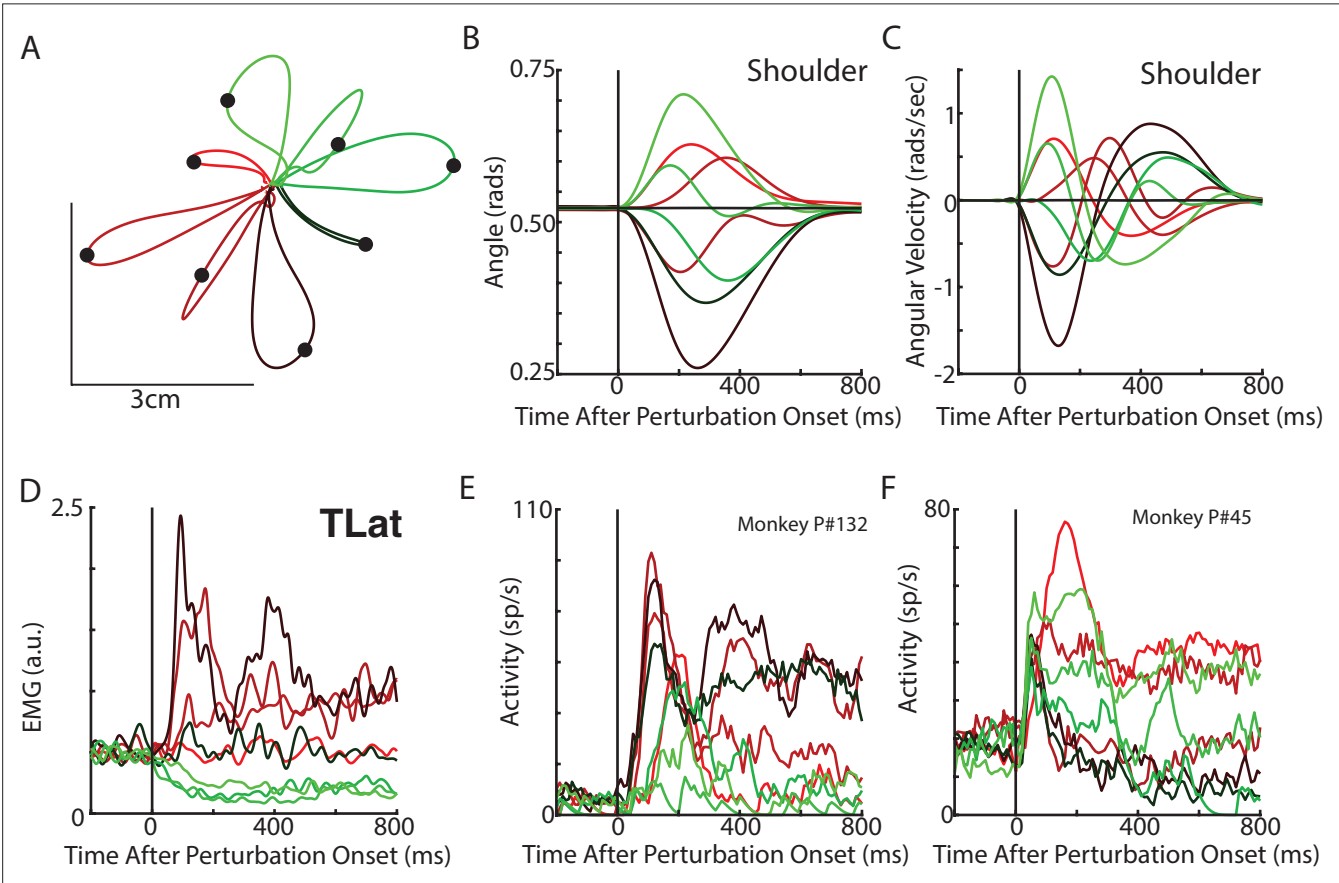

**Figure 4.** Posture perturbation task performed by monkeys. (**A**) Hand paths for Monkey P when mechanical loads were applied to its arm. (**B, C**) Shoulder angle and angular velocity aligned to the onset of the mechanical loads. (**D**) Recording from the lateral head of the triceps (elbow extensor) during the posture perturbation task. (**E, F**) Example neurons from motor cortex aligned to perturbation onset.

identical frequencies of rotation (*Figure 3—figure supplement 1A*). This indicates sensory feedback provided rich signals that could exhibit rotational dynamics identical to the network's dynamics.

## Motor and somatosensory cortex exhibit rotational dynamics while monkeys performed a posture perturbation task

Next, we examined if rotational dynamics exist in MC activity. We trained five monkeys to perform a similar posture perturbation task. The limb kinematics were qualitatively similar to the network with limb displacements of ~3 cm and hand reversal starting in 300–400 ms (*Figure 4A–C*). Muscle activity tended to be multi-phasic within the first 500 ms after the applied load and reached a steady state within 800 ms (*Figure 4D*). We also examined data from two previously collected monkeys performing a similar task using an endpoint manipulandum (data from *Chowdhury et al., 2020*). These monkeys also exhibited fast corrective movements to the load applied to the manipulandum (*Figure 3—figure supplement 2A-C*).

Neural activities were recorded using single electrodes (Monkeys P, A, and X) and chronic multi-electrode arrays (Monkeys Pu, M, H, and C). We observed MC responses tended to peak in <200 ms after the applied load and also exhibited steady-state activity (*Figure 4E–F*).

We pooled MC neurons across monkeys and then applied jPCA analysis (n=553). We found clear rotational dynamics in the top-2 jPC planes with frequencies of 1.3 Hz and 1.1 Hz for the first and second planes, respectively (*Figure 5A*). These planes also captured 63% of the variance from the neural population. In the third plane, we observed expansion-like dynamics similar to the third plane of the neural network (data not shown, 12% of variance). When we examined the fit qualities, we found the constrained and unconstrained dynamical systems had significant fits with an $R^2$ of 0.41 ($P$ < 0.001) and 0.50 (p<0.001), respectively (*Figure 5B*, blue lines, 'Group Pop.'). Similar results were found when we applied jPCA for each monkey. For Monkeys P, A, X, and Pu, we found population activities exhibited rotational dynamics in the top-2 jPC planes (*Figure 5—figure supplement 1A-D*, rotation frequency range: plane 1=2.4–1.6 Hz, plane 2=1.4–1.2 Hz). Significant fits were found for the constrained (*Figure 5B*; mean across monkeys $R^2$=0.45, p<0.01) and unconstrained dynamical systems (mean $R^2$=0.56, p<0.05). However, for Monkey M, we observed less rotational structure and more tangled trajectories in the top-2 jPC planes (*Figure 5—figure supplement 1E*). Fits for the constrained and unconstrained dynamical systems were still significant (constrained: p=0.003, unconstrained: p=0.002) but notably lower than for the other monkeys (constrained $R^2$=0.21, unconstrained $R^2$=0.32).

Rotational dynamics in MC during this task may reflect the inputs to MC from cortical areas upstream such as somatosensory and parietal areas (*Perich et al., 2020*). We explored this hypothesis by examining the population dynamics in somatosensory (S1, A2) and parietal (A5) cortices during this task. When neurons were pooled across monkeys (n=219), we observed clear rotational dynamics in the top-2 jPC planes with rotational frequencies of 1.7 Hz and 1.1 Hz (*Figure 5C*). Significant fits were found for the constrained (*Figure 5D*; $R^2$=0.49, p<0.001) and unconstrained ($R^2$=0.56, p<0.001) dynamical systems that were comparable to MC. Similar results were found when we applied jPCA for each monkey and cortical area separately (*Figure 5D*, *Figure 3—figure supplement 2D, E*, *Figure 5—figure supplement 2*).

Next, we examined the dynamics of the muscle activities and kinematic signals. We observed no rotational dynamics in the muscle activities for any of the monkeys (*Figure 5E*). We found the fits for the constrained and unconstrained dynamical systems were poor (Monkeys P/A/Pu/X: constrained: $R^2$=0.07/0.02/0.04/0.09, unconstrained: $R^2$=0.19/0.07/0.09/0.17) and were significantly worse than the down-sampled neural activity for all but Monkey P (probability values plotted in *Figure 5F*). In contrast, for the joint kinematics, we observed clear rotational dynamics with a rotation frequency of 1.4±0.1 Hz (across monkeys mean and SD; *Figure 5G*, *Figure 3—figure supplement 2F*). We found the fits for the constrained and unconstrained dynamical systems were good (constrained: $R^2$=0.53/0.47/0.38/0.54, unconstrained: $R^2$=0.64/0.61/0.45/0.62) and significantly greater than the down-sampled neural activity (probability values plotted in *Figure 5H* and *Figure 3—figure supplement 2G*). Finally, for each monkey, we also decoded M1's activity in each jPC plane using the joint kinematics and muscle activity and found the decoded activity was similar to M1's activity (*Figure 5—figure supplement 3*).

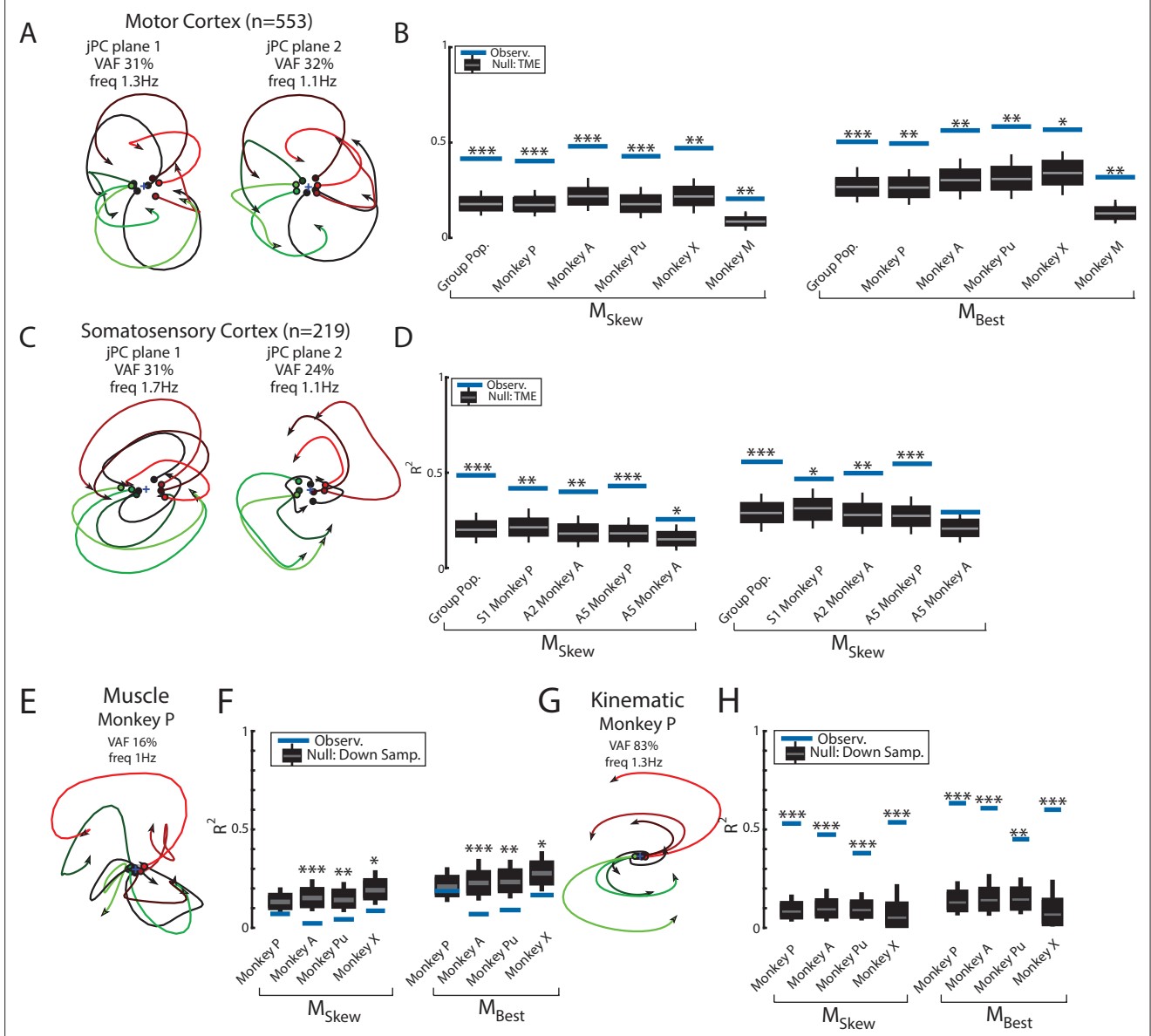

**Figure 5.** Population dynamics across motor and somatosensory cortex. (**A**) The top-2 jPC planes from activity recorded in motor cortex (MC) pooled across all monkeys. (**B**) Goodness of fits to the constrained ($M_{Skew}$ left) and unconstrained ($M_{Best}$ right) dynamical systems for MC activity for the pooled activity across monkeys (Group Pop.) and for each individual monkey. Null distributions were computed using tensor maximum entropy (TME). (**C, D**) same as (**A, B**) for somatosensory recordings. (**E**) The top jPC plane from muscle activity from Monkey P. (**F**) Goodness of fits to the muscle activity for the constrained and unconstrained dynamical systems for each monkey. (**G, H**) same as (**E, F**) for kinematic signals. (**B, D, F, H**) Gray bars denote the medians, the boxes denote the interquartile ranges, and the whiskers denote the 10th and 90th percentiles. *p<0.05, **p<0.01, ***p<0.001.

The online version of this article includes the following figure supplement(s) for figure 5:

**Figure supplement 1.** Population dynamics in motor cortex (MC) for individual monkeys.

**Figure supplement 2.** Population dynamics in somatosensory cortex for individual monkeys.

**Figure supplement 3.** Decoding M1 activity using kinematic and muscle activities.

## RNN exhibit rotational dynamics in the network activities and sensory feedback signals during delayed reach task

Rotational dynamics were first described in MC during a delayed reaching task (*Churchland et al., 2012*). We explored if our network also exhibited similar rotational dynamics by training it on a delayed center-out reaching task. The plant dynamics and network architecture were the same as

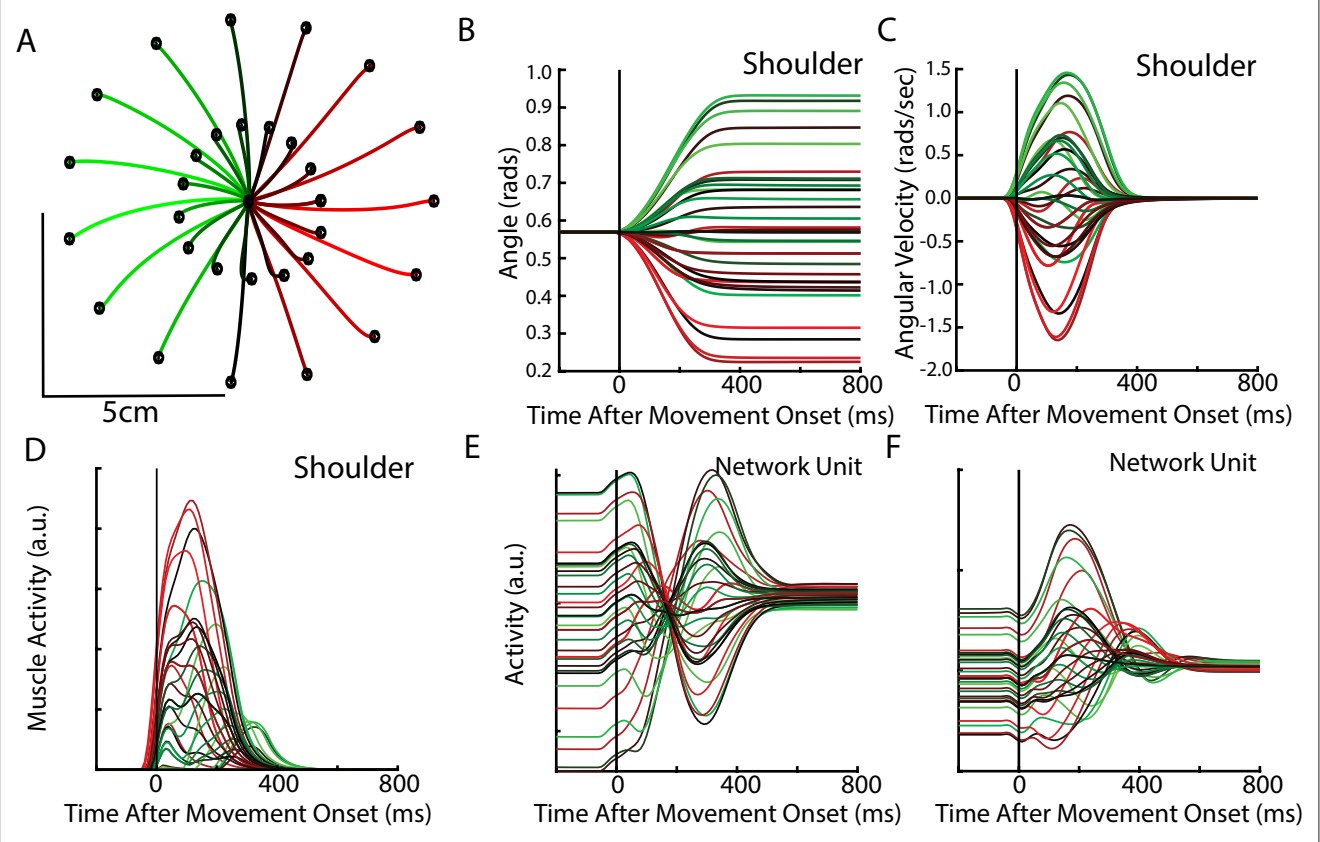

**Figure 6.** Delayed reach task by the network. (**A**) The hand paths by the model's arm from the starting position (center) to the different goal locations (black dots). Goals were placed 2 cm and 5 cm from the center location. (**B, C**) Shoulder angle and angular velocity aligned to movement onset. (**D**) Activity of the shoulder extensor aligned to GO cue onset. (**E, F**) The activities of two example units from the output layer of the network.

The online version of this article includes the following figure supplement(s) for figure 6:

**Figure supplement 1.** Population dynamics in somatosensory cortex during reaching from *Chowdhury et al., 2020*.

**Figure supplement 2.** Behavioral performance of the NO-REC network.

the posture task. However, the network was trained to maintain the limb at the starting location while the goal's location was provided as an input to the network (delay period). Following a variable time delay, a 'GO' cue was provided requiring the network to move the limb to the target location within ~500 ms. Note, the task-goal inputs to the network included the spatial location of the goal and GO cue (*Figure 1D*).

After optimization, the REC network was able to generate limb reaches toward radially located targets at displacements of 2 cm and 5 cm from the initial location (*Figure 6A*). Reaches had bell-shaped velocity profiles, that peaked roughly during the middle of the movement (*Figure 6B–C*). *Figure 6D* shows the activity of the shoulder extensor muscle during reaches to different target locations. *Figure 6E–F* show the diverse temporal profiles exhibited by units in the output layer of the network. The unit in *Figure 6E* has a stable response during the delay period when the target was present. After the 'GO' signal, the unit exhibits oscillatory activity with a change in the unit's preferred direction. The unit in *Figure 6F* largely maintains its preferred direction during the delay and movement periods.

We applied jPCA analysis to the output layer of the network and found clear rotational dynamics with rotational frequencies of 2.1 Hz and 1.1 Hz for the first and second planes, respectively (*Figure 7A*). These planes also captured 83% of the variance of the output-layer activity. When we examined the fit qualities, we found significant fits for the constrained and unconstrained dynamical systems with an $R^2$ of 0.70 (p<0.001) and 0.83 (p<0.001), respectively (*Figure 7B*). Note, the ratio between the $R^2$ for the constrained and unconstrained dynamical fits was 0.84, which is comparable to previous

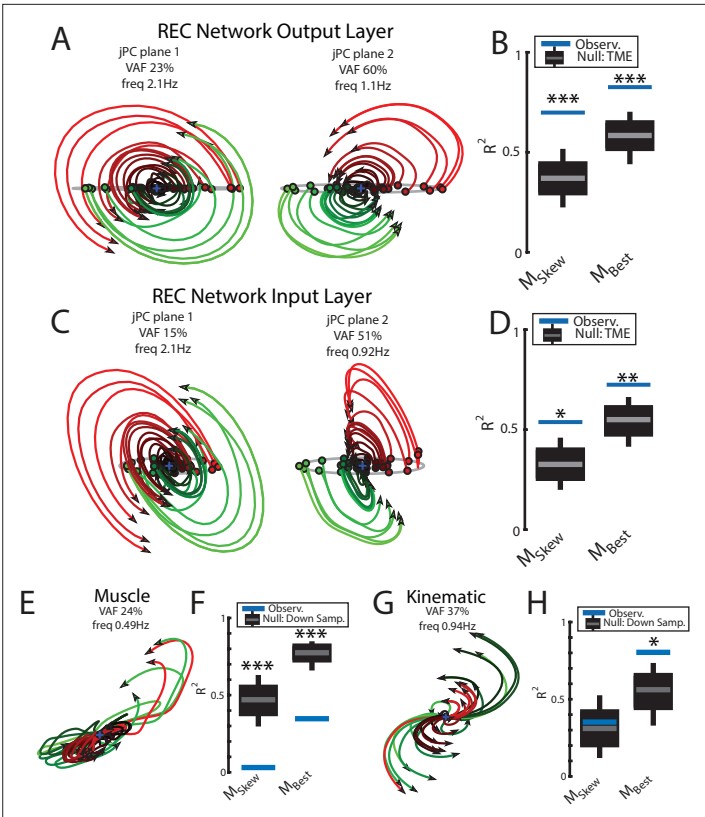

**Figure 7.** Population dynamics of the network during reaching. (**A**) The top-2 jPC planes from the output layer of the network during reaching. (**B**) Goodness of fits for the network activity to the constrained ($M_{Skew}$ left) and unconstrained ($M_{Best}$ right) dynamical systems. Null distributions were computed using tensor maximum entropy (TME). (**C, D**) same as (**A, B**) for the input layer of the network. (**E, F**) and (**G, H**) same as (**A, B**) except for the muscle activities and kinematic inputs into the network, respectively. Null distributions were computed from the down-sampled neural activity. (**B, D, F, H**) Gray bars denote the medians, the boxes denote the interquartile ranges, and the whiskers denote the 10th and 90th percentiles. *p<0.05, **p<0.01, ***p<0.001.

studies during reaching (*Churchland et al., 2012*) and indicates that the majority of the output layer's dynamics displayed rotational dynamics.

We also examined the input layer of the network and found essentially the same results as the output layer (*Figure 7C and D*). Clear rotational dynamics were present rotating at 2.1 Hz and 0.9 Hz in the top-2 planes, with significant fits for the constrained ($R^2$=0.54, p=0.01) and unconstrained ($R^2$=0.72, p=0.006) dynamical systems.

Next, we examined the dynamics of the muscle and kinematic signals. Similar to *Churchland et al., 2012*, we observed no rotational dynamics in the muscle activities (*Figure 7E and F*) and the fit for either dynamical system was significantly worse than the down-sampled network activity (constrained $R^2$=0.03, p<0.001; unconstrained $R^2$=0.35, p<0.001). In contrast, we observed rotational dynamics in the kinematic signals with a rotation frequency of 0.94 Hz (*Figure 7G and H*). We found the kinematic signals were better fit by both dynamical systems and were comparable to the down-sampled neural activity (constrained $R^2$=0.35, p=0.49; $R^2$=0.80, p=0.02). Further, when we predicted the output layer's activities using the combined sensory feedback (muscle, kinematics, GO cue, and static inputs), we again found the predicted activities were highly similar ($R^2$=0.99) to the output layer activities with virtually identical frequencies of rotation (*Figure 3—figure supplement 1B*).

## Somatosensory cortex exhibits rotational dynamics while monkeys performed delayed reaching task

Next, we explored if these dynamics were also present in somatosensory cortex during reaching, as previously observed in MC (*Churchland et al., 2012*). Monkeys H and C also completed a center-out

reaching task using a manipulandum and data were recorded from area 2 (data from *Chowdhury et al., 2020*; *Figure 6—figure supplement 1A*). Note, these monkeys made slightly slower reaches (~400 ms *Figure 6—figure supplement 1B*, C) than the reaches performed by the monkeys in *Churchland et al., 2012* as well as our model simulations (both ~300 ms).

We found clear rotational dynamics in area 2 with the top jPC plane having rotational frequencies of 1.0 Hz and 1.7 Hz for Monkeys H and C, respectively (*Figure 6—figure supplement 1D*). We also found significant fits for the constrained (*Figure 6—figure supplement 1E*, mean across monkeys $R^2$=0.51, p<0.001 both monkeys) and unconstrained ($R^2$=0.66, p<0.001) dynamical systems.

Examining the kinematics, we observed clear rotational dynamics in the top jPC plane with rotational frequencies of 1.3 Hz and 1.2 Hz for Monkeys H and C, respectively (*Figure 6—figure supplement 1F*). We also found significant fits for the constrained (*Figure 6—figure supplement 1G*, $R^2$=0.39, Monkey H p<0.001, Monkey C p=0.02) and unconstrained ($R^2$=0.51, Monkey H p<0.001, Monkey C p=0.01) dynamical systems.

## Neural networks without recurrent connections still exhibit rotational dynamics while performing posture and reaching tasks

A common assumption about rotational dynamics in MC is that they emerge from the intrinsic recurrent connections between neurons in MC. However, in our model, the sensory feedback into the network exhibited clear rotational dynamics that could contribute to the network's dynamics. Thus, we explored if networks trained to perform the posture perturbation task without the recurrent connections (input and output layers) also exhibit rotational dynamics (i.e., $\dot{r}(t) = f\left(s^*, s\left(t - \Delta\right)\right)$. Note, by removing the recurrent connections these networks can only generate time-varying outputs by exploiting the

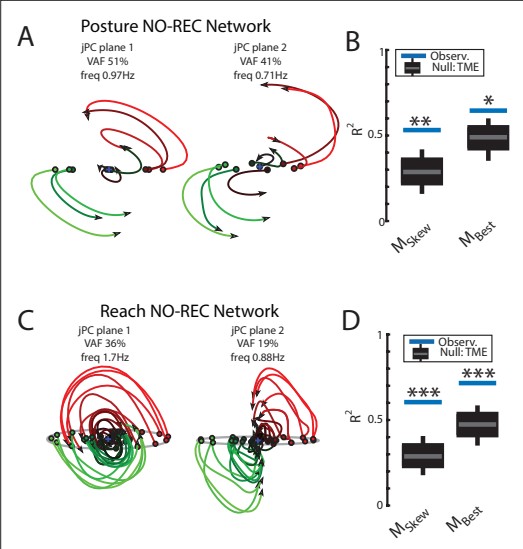

**Figure 8.** Population dynamics when trained without recurrent connections. Networks were trained to perform the posture and reaching tasks without the recurrent connections within the MC and input layers. (**A**) The top-2 jPC planes from the output layer of the network during the posture task. (**B**) Goodness of fits for the network activity to the constrained ($M_{Skew}$ left) and unconstrained ($M_{Best}$ right) dynamical systems. Null distributions were computed using tensor maximum entropy (TME). (**C, D**) same as (**A, B**) for the output layer of the network during the reaching task. (**C, D**) Gray bars denote the medians, the boxes denote the interquartile ranges, and the whiskers denote the 10th and 90th percentiles. *p<0.05, **p<0.01.

The online version of this article includes the following figure supplement(s) for figure 8:

**Figure supplement 1.** Analysis of REC network currents.

**Figure supplement 2.** REC networks perform better with continuous sensory feedback when encountering novel motor noise.

**Figure supplement 3.** Impact of sensory delays on rotational dynamics in network.

**Figure supplement 4.** Cartesian-based sensory feedback reduces rotational dynamics.

time-varying sensory inputs from the limb. We removed the recurrent connections in both the input and output layers of the network and optimized the network to perform the same posture task (NO-REC network). The network learned to bring the arm back to the central target when the external load was applied with kinematics similar to the REC network (*Figure 6—figure supplement 2A-C*) including similar displacement caused by the load (*Figure 6—figure supplement 2D*).

Examining the output-layer activity, we still observed clear rotational dynamics with rotational frequencies of 1.0 Hz and 0.74 Hz for the first and second planes, respectively (*Figure 8A*). These planes captured 92% of the variance of the network activity. When we examined the fit qualities, we found significant fits for the constrained dynamical system with an $R^2$ of 0.43 (*Figure 8B*, left; p=0.02), whereas for the unconstrained dynamical system, we found a fit with an $R^2$ of 0.54 but was

not significant (*Figure 8B*, right; p=0.3). As expected, output layer activities could be predicted from the sensory inputs with high accuracy (*Figure 3—figure supplement 1C*).

Finally, we examined if the rotational dynamics would also occur in a network without recurrent connections for the center-out reaching task (NO-REC). We found this network exhibited good control of the limb with qualitatively similar hand paths to the targets as the REC network during reaching (*Figure 6—figure supplement 2E,G*). However, the NO-REC network exhibited greater variability in the time to reach the goal as compared to the REC network (*Figure 6—figure supplement 2H*). Examining the output layer's dynamics, we observed rotational dynamics with rotational frequencies of 1.4 Hz and 0.85 Hz for the first and second planes, respectively (*Figure 8C*). These planes captured 82% of the variance of the network activity. When we examined the fit qualities, we found significant fits for the constrained dynamical system with an $R^2$ of 0.46 (*Figure 8D*, left; p=0.01), whereas for the unconstrained dynamical system, we found a fit with an $R^2$ of 0.56 but was not significant (*Figure 8D*, right; p=0.15). Again, output layer activities could be predicted from the sensory inputs with high accuracy (*Figure 3—figure supplement 1D*).

## Analysis of REC network's inputs reveal a strong bias Toward sensory inputs

For the NO-REC networks, the network dynamics are generated solely through sensory and task-goal feedback. It is less clear what the relative importance of sensory inputs is in the REC network due to dynamics that could also be generated through intrinsic recurrent connections. Here, our objective is to quantify the relative importance of sensory inputs and inputs from intrinsic recurrent connections. For each neural unit in the input layer of the REC network, we examined their synaptic weights for the sensory feedback connections and intrinsic recurrent connections from neurons in the layer (*Figure 8—figure supplement 1A*). We computed the ratio between the norm of the local recurrent connections and the sensory feedback connections for each individual neuron (*Figure 8—figure supplement 1B, C*). The resulting distribution was slightly larger than one indicating weights were larger for the recurrent connections than the feedback connections. However, examining the relative synaptic weights can be misleading as the total contribution of an input to a neuron's response is the product of the weight with the activity of the input signal (*Figure 8—figure supplement 1A*, E[s] and E[r]). Thus a more appropriate comparison is to compare the currents (W*E[·]) generated from intrinsic and sensory sources. *Figure 8—figure supplement 1E, F* show the ratio of the intrinsic currents with the sensory currents across neurons. The distributions are centered near 0.5 indicating that the sensory contribution is ~ 2× larger than the recurrent contribution across neurons. Thus, sensory inputs had a substantial impact in generating the dynamics in the REC networks. However, we caution interpreting these results in the context of a biological system as many factors not modeled will likely contribute to the relative weighting of intrinsic dynamics and sensory feedback.

## Sensory feedback improves network performance when encountering motor noise

The above results show that networks that receive sensory feedback and trained to perform the perturbation posture and reach tasks display rotational dynamics. Rotational dynamics are also possible in networks that are purely driven by intrinsic recurrent connectivity (*Hennequin et al., 2014*; *Michaels et al., 2016*; *Sussillo et al., 2015*). However, the motor system is impacted by noise at different stages including noise that corrupts muscle activation in a signal-dependent manner. Sensory feedback becomes indispensable in correcting errors from an intended movement. To demonstrate the robustness of feedback networks over autonomous networks for dealing with motor noise, we trained separate instances of the REC networks in the posture task where the network was trained to reach the original location following a perturbation with zero velocity. Critically, we included networks that either received continuous sensory feedback or received a pulse of sensory feedback for the first 200 ms after the perturbation (*Figure 8—figure supplement 2A*). Note, we could not include a fully autonomous network as the cue to correct for the mechanical load is provided only through sensory feedback in our model. After training, we compared how well the networks performed when motor noise was added to the motor commands (note: motor noise was not present during the training of the network). *Figure 8—figure supplement 2B* shows a clear reduction in the endpoint position error of the hand (kinematic error) for the continuous feedback network as compared to the pulse feedback

networks when the variance of the motor noise was 80% of the motor commands. Across motor noise levels, the continuous feedback network outperformed the pulse feedback (*Figure 8—figure supplement 2C*).

Similar results were found during the reaching task (*Figure 8—figure supplement 2D-F*). Note, here we compared the continuous feedback network with a fully autonomous network (i.e., no sensory feedback) as the information about the reach target and when to initiate the reach is provided by the task-goal and GO cue inputs. The continuous feedback network resulted in smaller endpoint position error of the hand than the fully autonomous network across motor noise levels (*Figure 8—figure supplement 2E, F*).

## Increase in sensory feedback delays has a small attenuating impact on rotational frequency

Next, we explored how sensory feedback delays impacted the network's dynamics. We considered three different feedback delays (0 ms, 50 ms, and 100 ms) and trained up to five separate networks with random initialization for each delay. In the posture task, we did not find substantial change in the fit qualities for the constrained and unconstrained dynamical systems for the REC network (*Figure 8—figure supplement 3A*, top row; mean value across initializations for 0/50/100 ms delay: constrained 0.38/0.47/0.44; unconstrained 0.67/0.81/0.83) and a trend toward increased fit quality for the unconstrained dynamical system for the NO-REC network (*Figure 8—figure supplement 3B*, top row; constrained 0.34/0.45/0.44, unconstrained 0.47/0.57/0.74). Rotational frequencies for the first jPC plane tended to be slower with greater sensory delay for both the REC and NO-REC networks (*Figure 8—figure supplement 3A, B* bottom rows; frequency 1.60/1.57/1.42 Hz). Note, we adjusted our analysis window in accordance with the delay as changes in the network's dynamics could only begin when sensory feedback about the limb had reached the network. Otherwise, reduction in fit qualities and rotational frequencies with greater delay would simply reflect that the network was still in the pre-perturbation state without knowledge of which load had been applied for a portion of the analysis time window.

Similar trends were observed during reaching for the REC network (*Figure 8—figure supplement 3C*). There was a trend toward a lower fit quality for the constrained dynamical system (constrained 0.72/0.61/0.58, unconstrained 0.82/0.72/0.79) with the longer delay and a small reduction in the rotational frequency of the first jPC plane (1.90/1.60/1.47 Hz). For the NO-REC network, we saw fit qualities (constrained 0.69/0.59/0.42, unconstrained 0.82/0.72/0.57) and rotational frequencies tended to be reduced with greater sensory delays (*Figure 8—figure supplement 3D*, frequency 1.52/1.45/1.12 Hz). Note, here we did not adjust our analysis window in accordance with delay as the GO cue initiates when the network state begins to change and its arrival to the network was unaffected by the sensory feedback from the limb.

## Neural networks with cartesian-based rotational dynamics exhibit rotational dynamics

Next, we determined how the representation of the sensory feedback signals impacted the network dynamics. In a set of simulations, we trained up networks where sensory feedback of the limb's kinematics were the two-dimensional position and velocity of the hand in cartesian coordinates rather than angle and angular velocity of the joints. Networks were able to perform both tasks well with performance comparable to the networks with joint-based sensory feedback (data not shown).

In the posture task, there was a noticeable reduction in the REC network's rotational dynamics (*Figure 8—figure supplement 4A*). Fit quality for the constrained and unconstrained dynamical systems were 0.2 and 0.62, respectively (*Figure 8—figure supplement 4B*, left), which were noticeably smaller than for the joint-based feedback (*Figure 3B*, constrained $R^2$=0.55, unconstrained $R^2$=0.83). Interestingly, when we examined the kinematic signals in the cartesian reference frame, we still found strong fits for both dynamical systems (*Figure 8—figure supplement 4E*, constrained $R^2$=0.67, unconstrained $R^2$=0.79).

One possible reason for the reduction in rotational dynamics might be due to initializing the network weights using a uniform distribution with a range from $\pm 1/\sqrt{N}$ where N is the number of neural units. In contrast, previous studies have initialized the network weights using a Gaussian distribution with standard deviation equal to $g/\sqrt{N}$ where g is constant larger than 1. This alternative

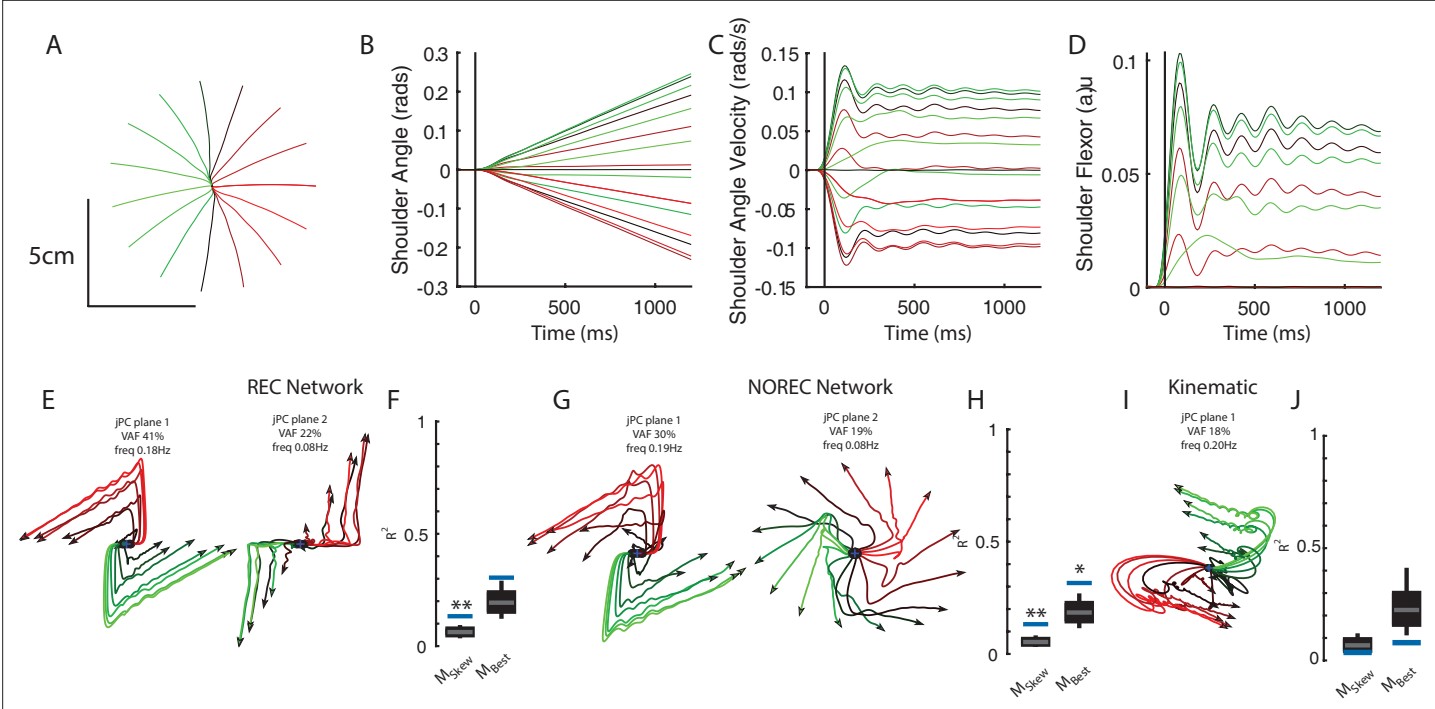

**Figure 9.** Networks trained on constant velocity tracking task exhibit dynamics that are less rotational. (**A**) Hand paths of the model performing the task. The limb started at the center and followed 1 of 16 trajectories in the radial direction. (**B**) and (**C**) Shoulder angle and angular velocity during the task. (**D**) Shoulder flexor muscle during task. (**E**) Activities for the output layer of the REC network for the top-2 jPC planes. Activity was examined from the start of movement till the end of the trial (~1.2 s). (**F**) Fits for the constrained and unconstrained dynamical systems. (**G, H**) same as (**E, F**) except for the NO-REC network. *p<0.05, **p<0.01.

initialization scheme encourages strong intrinsic dynamics often needed for autonomous RNN models (*Sussillo et al., 2015*). We found networks initialized with this method exhibited stronger rotational dynamics with fits to the constrained and unconstrained dynamical systems of 0.5 and 0.88, respectively (*Figure 8—figure supplement 4C, D*).

When examining the reaching task, we found similar results (*Figure 8—figure supplement 4F-K*). When initialized with a uniform distribution, fit quality for the constrained and unconstrained dynamical systems were 0.4 and 0.77, respectively (*Figure 8—figure supplement 4F,G*), which were smaller than for the joint-based feedback (*Figure 7B*, constrained $R^2$=0.7, unconstrained $R^2$=0.83). Qualitatively, the dynamics were different when the network was initialized with a Gaussian distribution (*Figure 8—figure supplement 4H*); however, fit qualities were comparable between the two initialization methods (*Figure 8—figure supplement 4I*). There was also a noticeable reduction in the fit quality for the kinematic signals in the cartesian reference frame, particularly for the constrained dynamical system (*Figure 8—figure supplement 4K*, constrained $R^2$=0.36, unconstrained $R^2$=0.77).

## Tracking a constant velocity target diminishes rotational dynamics in the neural network

Finally, we wanted to observe if the emergence of rotational dynamics in networks generalized to other behavioral tasks. In particular, we focused on a tracking task similar to the center-out reaching task but where the network now tracks the movement of a target traveling at a constant velocity in the radial direction. Hence, unlike the reach and posture tasks, the motion of the limb does not stop in the tracking task. Note, the target's position was provided to the network as a task-goal input (*Figure 1E*), while both joint angle and angular velocity are provided as delayed sensory feedback. Both REC and NO-REC networks were able to complete the task with straight hand trajectories (*Figure 9A*, REC network shown) with limb motion that had constant velocity (*Figure 9B and C*). The accompanying muscle activity initially increased to accelerate the limb from rest and reached a steady state of near-constant output over the remainder of the trial (*Figure 9D*).

Examining the output layer activities of the REC network, we observed dynamics that were less rotational than we observed for the perturbation posture and reaching task (*Figure 9E*). Note, we analyzed activity from the start of the movement to the end of the trial (~1.7 s). The top-2 jPC planes captured 63% of the total variance and had rotational frequencies of 0.18 Hz and 0.08 Hz (*Figure 9E*). Fit qualities to the constrained and unconstrained dynamical system were also poor with $R^2$ of 0.13 and 0.3, respectively (*Figure 9F*). However, the results for the constrained dynamical system were still significant according to the TME null distribution as this distribution is dependent on the statistics of the observed network activities (see *Elsayed and Cunningham, 2017*). Similar results were found when we observed the NO-REC network (*Figure 9G*) with fits to the constrained and unconstrained dynamical systems of 0.13 and 0.32, respectively (*Figure 9H*). Finally, applying jPCA to the kinematic signals revealed poor fits for the constrained and unconstrained dynamical systems (*Figure 9I and J*, constrained $R^2=0.04$, unconstrained $R^2=0.08$). Note, initializing the network weights using a Gaussian distribution yielded similar findings (data not shown).

## Discussion

The present study highlights how neural network models with sensory feedback and intrinsic recurrent connections exhibit rotational dynamics in the network activities and in the sensory feedback from the limb, but not in muscle activities. These rotational dynamics were observed for a postural perturbation and a delayed reaching task, and critically, even without intrinsic recurrent connections in the model. Similar tasks performed by monkeys also illustrate rotational dynamics not only in MC, but also in somatosensory areas and likely in sensory feedback signals related to joint motion. Thus, rotational dynamics are a characteristic that is present throughout the sensorimotor system, just not for muscles.

Churchland et al. found rotational dynamics in MC during reaching. These dynamics were well described by an autonomous dynamical system ($\dot{X} = M \cdot X$ where the system evolves in time ($\dot{X}$) based solely on its current state using recurrent dynamics ($M \cdot X$)). Furthermore, *Sussillo et al., 2015* found rotational dynamics in RNNs trained to generate the same patterns of muscle activity observed during reaching. Critically, similar rotational dynamics were generated whether the networks received relatively simple inputs or inputs that simulated online sensory feedback. Although both studies leave open the question about how these recurrent dynamics are generated, whether it is through intrinsic (e.g., local synapses) or extrinsic factors (e.g., sensory feedback), these studies have been interpreted as evidence that these dynamics are generated from intrinsic connectivity in MC (*Bizzi and Ajemian, 2020*; *Collinger et al., 2018*; *Diedrichsen and Kornysheva, 2015*; *Kalaska, 2019*; *Khanna et al., 2021*; *Pandarinath et al., 2018a*; *Perich et al., 2020*; *Rouse et al., 2018*; *Suminski et al., 2015*; *Suresh et al., 2020*; *Vyas et al., 2020*).

The present study cannot directly refute this possibility, but it does provide several observations that clearly do not fit with this interpretation. First, we observed rotational dynamics in sensory feedback from the limb. Previous RNNs models of MC only used EMG-like signals for sensory feedback (*Sussillo et al., 2015*). Given muscle activity does not show rotational dynamics, it is perhaps not surprising that EMG-like feedback signals also show no rotational dynamics. However, primary and secondary afferents are critical sources of sensory feedback for limb control and their activity correlates with muscle length and change in that length (*Cheney and Preston, 1976*; *Edin and Vallbo, 1990*; *Loeb, 1984*). Our model and analysis of experimental data quantified joint angular position and velocity as a proxy of these sensory signals and found that they displayed rotational dynamics. This emerges during the posture perturbation and reaching tasks due in part to the phase offset between the joint position and velocity as changes first occur in the velocity followed by position (see pendulum example *Pandarinath et al., 2018b* also *DeWolf et al., 2016*; *Susilaradeya et al., 2019*). This phase offset is maintained across reach directions and gives rise to the orderly rotational dynamics observed in kinematic signals (*DeWolf et al., 2016*; *Pandarinath et al., 2018a*; *Susilaradeya et al., 2019*; *Vyas et al., 2020*). Furthermore, the tracking task disrupted this phase relationship, and thus, the rotational dynamics were substantively reduced in the network models.

Second, neural network models displayed rotational dynamics even when there were no intrinsic recurrent connections (NO-REC). Instead, these networks inherited their dynamics solely from the sensory inputs from the limb. This suggests that rotational dynamics in MC may reflect internal dynamics, system inputs, or any weighted combination of the two.

Third, rotational dynamics were observed not only in MC, but also in somatosensory cortex during the posture perturbation and reaching tasks. Rotational dynamics were observed in S1 (areas 3 a and 1), A2 and A5, and a recent study has even identified rotational dynamics in the rostral areas of posterior parietal cortex (V6a, *Diomedi et al., 2021*). These areas reflect important components of frontoparietal circuits involved in the planning and execution of arm motor function (*Chowdhury et al., 2020*; *Kalaska, 1996*; *Kalaska et al., 1990*; *Omrani et al., 2016*; *Takei et al., 2021*). Thus, rotational dynamics are observed throughout frontoparietal circuits and likely in sensory feedback from the limb.

Importantly, findings of rotational dynamics in cortical circuits are not trivial. Activity in the supplementary motor area does not exhibit rotational dynamics during reaching (*Lara et al., 2018*). The hand area of MC also does not exhibit rotational dynamics during grasping-only behaviour (*Suresh et al., 2020*), though it does exhibit rotational dynamics during reach-to-grasp (*Abbaspourazad et al., 2021*; *Rouse and Schieber, 2018*; *Vaidya et al., 2015*) which may reflect the reaching component of the behaviour. More broadly there is a growing body of work characterizing cortical neural dynamics across different behavioral tasks which have revealed rotational (*Abbaspourazad et al., 2021*; *Aoi et al., 2020*; *Gao et al., 2016*; *Kao et al., 2015*; *Libby and Buschman, 2021*; *Remington et al., 2018*; *Sani et al., 2021*; *Sohn et al., 2019*; *Stavisky et al., 2019*; *Vaidya et al., 2015*), helical (*Russo et al., 2020*), stationary (*Machens et al., 2010*), and ramping dynamics (*Finkelstein et al., 2021*; *Kaufman et al., 2016*; *Machens et al., 2010*) and these dynamics appear to support various classes of computations. Thus, finding of rotational dynamics across the fronto-parietal circuit in the present study was not trivial.

One feature not captured by our model is that complex multi-phasic activity patterns precede movement onset by 100–150 ms. Obviously, sensory feedback of the movement cannot play a role in generating this early activity and must occur through internal processing including inputs from other brain regions (*Sauerbrei et al., 2020*). Given limb sensory feedback reaches cortex in 20 ms (*Evarts and Tanji, 1976*; *Pruszynski et al., 2011*; *Wolpaw, 1980*), our results suggest that sensory feedback is likely to contribute heavily to MC dynamics during movement.

Behavioral-level models also highlight a critical role for sensory feedback for motor control. OFC has been influential as a normative model of voluntary control for almost 20 years (*Scott, 2004*; *Todorov and Jordan, 2002*). These types of controllers include two basic processes. First, state estimation where the present state of the body is optimally calculated from various sensory signals as well as from internal feedback generated using forward models. Second, a control policy uses this state estimate to generate motor commands to move the limb to a behavioral goal. These models predict many features of our motor system including that it is highly variable but also successful, and the ability to exploit redundancy while attaining a goal reflecting an interplay between kinematic errors and goal-directed corrections (*Diedrichsen, 2007*; *Diedrichsen, 2007*; *Knill et al., 2011*; *Liu and Todorov, 2007*; *Nashed et al., 2012*; *Nashed et al., 2014*; *Scott, 2016*; *Trommershäuser et al., 2005*). A large body of literature highlights that goal-directed motor corrections to mechanical disturbances can occur in ~60 ms and involve a transcortical pathway through MC (*Matthews, 1991*; *Scott, 2004*; *Scott, 2012*). These observations point to the importance of sensory feedback processing as a continuous rather than an intermittent process providing a continuous stream of input to brain circuits to guide and control motor actions (*Crevecoeur and Kurtzer, 2018*).

However, it is important to stress that intrinsic processing during motor actions is also required for OFC. Optimal feedback controllers relay an efference copy of the motor commands to circuits involved with state estimation that estimate the future state of the limb using an internal model. This estimate is combined with sensory feedback and sent back to circuits involved with implementing the control policy (i.e., MC), and thus, creating a recurrent loop. Optimal feedback controllers also generate time-varying feedback gains that map the limb estimate to the desired motor commands (*Dimitriou et al., 2013*; *Liu and Todorov, 2007*). These feedback gains must be generated independent of the current limb estimate, and thus, requires dynamics that are generated intrinsically. Although in theory, this intrinsic processing could be done by local synaptic connections in MC, it is likely that intrinsic processing inside feedback loops between cortical and subcortical areas such as the cortico-cerebellar-cortical loops play an important role.

Thus, from the OFC perspective, the rotational dynamics in MC are generated by a number of factors including local intrinsic connections, inputs from other brain areas, and inputs from sensory feedback. Teasing apart how these factors combine and give rise to the observable dynamics remains

an important open question. Recent studies have suggested that MC uses an initial planning stage when processing visual feedback during movement. *Stavisky et al., 2017* showed that the initial visual feedback response to a shift in hand position during reaching may be transiently isolated from the activity associated with generating motor output. However, as we show here, this latter activity may still reflect sensory and internal feedback. Similarly, *Ames et al., 2019* showed that jumping the location of the goal during reaching to a new location generated activity patterns that were similar to the patterns generated when planning a separate reach to the new goal's location. This planning stage may reflect an update to the control policy given the visual error, resembling model predictive control (*Dimitriou et al., 2013*), and it remains an open question if these feedback responses to systematic errors (visual shift or mechanical load) evoke the same activity patterns in MC as motor noise (*Crevecoeur et al., 2012*).

## Materials and methods
### Two-link arm model
We constructed a two-link model of the upper arm as detailed in *Lillicrap and Scott, 2013*. The model was constrained to move in a horizontal two-dimensional plane and incorporated arm geometry and inter-segmental dynamics. The dynamics of the limb were governed by

$$\mathbf{x}_{t+1} = f\left(\mathbf{x}_t, \, \boldsymbol{\Gamma}_t\right) \tag{1}$$

where ' $\mathbf{x}_t$ ' is the vector state of the arm at time 't' and was composed of the angular positions and velocities of the elbow and shoulder joints $\left[\theta_{elb}, \, \theta_{sho}, \, \dot{\theta}_{elb}, \, \dot{\theta}_{sho}\right]$. ' $\boldsymbol{\Gamma}_t$ ' is the two-dimensional vector of torques applied to the shoulder and elbow joints at time 't'. We incorporated six-lumped muscle actuators that moved the arm, which included four mono-articular and two bi-articular muscles. These muscles received input from the neural network and exhibited force-length and force-velocity dependent activation properties (*Brown et al., 1999*). Muscle forces ($\mathbf{m}_t$) were converted to joint torques ($\boldsymbol{\Gamma}_t$) by computing the product between each muscle's force output with their respective moment arm. The parameters for the arm dynamics, moment-arm matrix and the muscle force-length/velocity (F-L/V) properties were drawn from the literature (*Brown et al., 1999*; *Cheng et al., 2000*; *Graham and Scott, 2003*). The continuous arm dynamics were discretized and solved using Euler's integration with a time step (dt) of 10 ms.

### Network description
We used a RNN composed of two layers to control the arm model. These layers are referred to as '**h**' and '**o**' as a shorthand for hidden layer (the first layer) and output layer (the second layer) in artificial neural networks (*Figure 1B*). Both layers had recurrent connections between units within each layer and all units had leaky-integration properties and a standard sigmoid activation function.

The first layer received inputs composed of a step signal representing the desired joint state ($\mathbf{x}_t^*$), delayed ($\Delta$= 50 ms) state feedback from the arm ($\mathbf{x}_{t-\Delta}$, joint angles and angular velocities) and delayed muscle activations ($\mathbf{m}_{t-\Delta}$). For the reaching task we also included a condition-independent binary 'GO' cue to indicate when the network should initiate movement. This signal was applied as a step function smoothed with a 20 ms s.d. Gaussian kernel (high indicates hold command, low indicates move command). For brevity, let us denote all of the external inputs to the first layer of the neural network at time 't' as $\mathbf{I}_t$ . The dynamics of the first layer (referred to as input layer) were governed by

$$\mathbf{h}_{t+1} = \left(1 - l_n\right)\mathbf{h}_t + l_n \tanh\left(\mathbf{W}_{sh}\mathbf{I}_t + \mathbf{W}_{hh}\mathbf{h}_t + \mathbf{b}_h\right) \tag{2}$$

where $\mathbf{h}_t$ is the vector of unit activities for the input layer, ' $l_n$ ' is the ratio between the simulation time-step (dt) and the time-constant of the network units ($\tau_n$), hence $l_n = dt/\tau_n$ . $\mathbf{W}_{sh}$ is the weight matrix that maps the inputs to the activities of the input layer, $\mathbf{W}_{hh}$ is the weight matrix for the recurrent connections between units in the input layer, and $\mathbf{b}_h$ is the bias (or baseline) for the first layer activities.

The second layer (output layer) received input from the input layer and its dynamics were governed by

$$\mathbf{o}_{t+1} = (1 - l_n)\,\mathbf{o}_t + l_n \tanh\left(\mathbf{W}_{ho}\mathbf{h}_t + \mathbf{W}_{oo}\mathbf{o}_t + \mathbf{b}_o\right) \tag{3}$$

where $\mathbf{o}_{t+1}$ is the vector of unit activities for the output layer, $\mathbf{W}_{ho}$ is the weight matrix that maps the input layer activities to the output layer activities, $\mathbf{W}_{oo}$ is the weight matrix for the recurrent connections between units in the output layer, and $\mathbf{b}_o$ is the bias (or baseline) for the output layer activities.

The output layer provides inputs to the six muscles used to control the limb. The muscle activities ($\mathbf{m}_t$) were governed by,

$$\mathbf{m}_{t+1} = (1 - l_m) \cdot \mathbf{m}_t + l_m \cdot \left[\mathbf{W}_{ou}\mathbf{o}_t\right]^+ \tag{4}$$

$\mathbf{W}_{ou}$ is the weight matrix that maps the activities in the output layer to the lumped muscle actuators, and $l_m$ is the time constant for the muscle given by, $l_m = dt/\tau_m$ .

We also examined networks where we removed the recurrent connections from each layer by setting $\mathbf{W}_{hh}$, $\mathbf{W}_{oo}$ to zero for the entire simulation and optimization while also removing the $(1 - l_n)\,\mathbf{h}_t$ term in *equation 2* and the $(1 - l_n)\,\mathbf{o}_t$ term in *equation 3* (NO-REC networks).

Note, although we used a two-layer network, similar results were found using a one-layer network, however, the training time needed to converge to a solution tended to be longer in the one-layer network.

For all simulations, the input and output layers were composed of N = 500 units each and the time constants of network units ($\tau_n$) and muscle units ($\tau_m$) were 20 ms and 50 ms, respectively. The weight matrices were initialized from a uniform distribution between $\left[-\frac{1}{\sqrt{N}_{inp}},\ \frac{1}{\sqrt{N}_{inp}}\right]$. $N_{inp}$ denotes the length of the rows (or the total number of incoming connections onto a neuron in a given layer) in a given weight matrix. All the bias vectors $\left[\mathbf{b}_h,\ \mathbf{b}_o\right]$ were initialized to 0.

## Choice of sensory inputs into network

Our model receives delayed sensory feedback from the periphery composed of the angles and angular velocities of the joints as well as the muscle activities. We think these are reasonable inputs into the network based on the known properties of proprioceptors. Activity of muscle spindles is known to signal muscle length and velocity (*Cheney and Preston, 1976*; *Edin and Vallbo, 1990*; *Loeb, 1984*), which could be used to form an estimate of joint angle and angular velocity (*Scott and Loeb, 1994*). Activity of Golgi tendon organs signal muscle tension and correlate with muscle activity (*Houk and Henneman, 1967*; *Nichols, 2018*; *Prochazka and Wand, 1980*).

## Task descriptions

We trained the network to perform a posture perturbation task similar to our previous studies (*Heming et al., 2019*; *Omrani et al., 2014*; *Pruszynski et al., 2014*). The network was required to keep the arm at a desired position while the limb was displaced by loads applied to the shoulder and elbow joints. Eight torques (of magnitude 0.2 N m) were used consisting of elbow flexion (EF), elbow extension (EE), shoulder flexion (SF), shoulder extension (SE), and the four multi-joint torques (SF+EF, SF+EE, SE+EF, SE+EE). Importantly, the network did not receive any explicit information on the direction of the applied load and had to use the delayed sensory feedback to produce appropriate compensation.

We also trained separate instances of the network to perform a delayed center-out reach task that required the network to hold the arm at a starting position for 500 ms. Afterwards, a GO cue appeared signaling the network to move to the target within 500 ms. We had the network reach to 32 different targets spaced radially around the starting position with half of the targets located 2 cm away from the starting position, and the remaining half were placed 5 cm away from the starting position. The network then had to hold at the reach target for the remainder of the trial (~500 ms). Note, for our simulations, we used a fixed time delay (represented by the GO signal) for when the network should initiate a reach to decrease optimization time. Simulations with a variable delay yielded virtually the same results.

For the tracking task, we modified the center-out reaching task by introducing a target that moved at a constant velocity, in contrast to reaching the target within 500 ms. We had the network perform constant speed tracking toward 15 different targets spaced radially around the starting position located 5 cm away from the starting position. In contrast to the center-out reaching task, we did not provide the network with any explicit GO cue to signal the network to start movement. Instead, the

network initiated movement when the target started moving after an initial rest period. The location of the target was provided as a ramping input to the network.

## Network optimization

For optimizing the networks, we defined the loss function (l) over a given trial (i) as

$$l^i = \sum_{t=0}^{T} \left\| \mathbf{x}_t^i - \mathbf{x}_t^{*i} \right\|^2 + \alpha \left\| \mathbf{m}_t^i \right\|^2 + \beta \left\| \mathbf{h}_t^i \right\|^2 + \gamma \left\| \mathbf{o}_t^i \right\|^2 \tag{5}$$

where $\alpha$, $\beta$, $\gamma$ are penalization weights. The first term of the loss function is the vector norm between the desired limb kinematic state ($\mathbf{x}_t^{*i}$) and the current limb kinematic state ($\mathbf{x}_t^i$). The second term penalizes the total muscle activity, and the third and fourth terms penalize high network activities for the first and second layers, respectively.

In the posture perturbation task, the desired limb state was static invariant to the direction of external torques, and the kinematic term considered the norm of the difference between the desired state of the arm and the actual state 1000 ms after the time of load application. In the reach task, the desired limb state was defined as the location of the reach target on that trial and the kinematic error was penalized 500 ms after the GO cue was presented. Essentially, our cost formulation for both posture and reach tasks evaluate the end kinematic error, in both arriving accurately to the spatial target (within 1000 ms in posture task, and 500 ms in reach task) and then subsequently holding the arm in that location with zero velocity for some time. Similar to the posture task, the muscle and network activities were penalized during the entire reach task. Notably, in both reach and posture tasks, we also penalized any kinematic deviation from the starting home location during the initial HOLD period (i.e., the time period before the presentation of GO cue in reach task, and before the time application of load in posture task). For the tracking task, the cost function penalized differences between the moving target's position and the arm's position and penalized deviations in the hand velocity from the target's velocity throughout the movement.

The network parameters were determined as an optimal solution that minimizes the total cost 'J' from summing the individual trial loss functions across different movement conditions (C) (i.e., the nine load combinations in the posture task or 32 target locations in the reach task), and across the task duration (T).

$$J = \frac{1}{2 \cdot C \cdot T} \sum_{i=1}^{C} l^i \tag{6}$$

We optimized the network by applying back-propagation through time (**Werbos, 1990**). This requires us to compute the cost-gradient ($\frac{\partial J}{\partial W}$) with respect to the adjustable network parameters $W = \left[ W_{sh}, W_{hh}, W_{ho}, W_{oo}, W_{ou}, b_h, b_o \right]$. Since, the total cost depends upon the kinematic state of the arm ($x_t$), the optimization problem involves calculating the Jacobian of the arm dynamics ($\frac{\partial x_t}{\partial u_t}$) at each time-step, as presented in **Stroeve, 1998**. Our simulations were implemented in Python and PyTorch machine learning library (**Paszke et al., 2017**). Optimization was performed using the Adam algorithm (**Kingma and Ba, 2017**) and performed until the network generated successful limb trajectories and the error had decreased to a small, constant valuer (approx. 1e−4) for at least 500 epochs. For all the simulations, the hyper-parameters were fixed at $\alpha$=1e−4/1e−3, $\beta$=1e−5/1e−6, and $\gamma$=1e−5/1e−6; although comparable network solutions were obtained for a broad range of these hyper-parameter values. Note, in the posture task, during a delayed period before the application of any load, the muscle activities were penalized with a higher $\alpha$=1e−2 to ensure that the muscles were not active by default at a higher baseline to counter-act the upcoming load.

### Neural recordings

We analyzed neural activity from fronto-parietal areas when monkeys performed a posture perturbation task that had been previously collected (**Chowdhury et al., 2020**; **Heming et al., 2019**; **Omrani et al., 2014**; **Omrani et al., 2016**; **Pruszynski et al., 2014**). Briefly, Monkeys P, A, X, Pu, and M had their arms placed in a robotic exoskeleton that restricted the animal's movements to motion of the shoulder and elbow joints in a two-dimensional horizontal plane. These animals performed almost the exact same posture perturbation task as the network. However, different load magnitudes were used for each monkey depending on their physical capabilities (Monkeys P, X=0.2 Nm, A=0.4 N m,

Pu=0.2 Nm, and M=0.34 N m). Also, for some recordings in Monkeys P, X, and M, the load was removed 300 ms after it was applied. Given that we were interested in the earliest feedback response, we included these recordings. Data for Monkeys H and C were from *Chowdhury et al., 2020* where the monkeys performed a similar task using a robotic manipulandum and where 2 N forces were applied to the manipulandum that lasted 125 ms (*London and Miller, 2013*).

Monkeys H and C also performed a delayed center-out reaching task (*Chowdhury et al., 2020*; *London and Miller, 2013*). Goal targets were arranged radially around the starting position at a distance of 12.5 cm. For Monkeys H and C, eight and four different goal locations were used, respectively. After the delay period, the monkeys had to reach for the goal location within ~2 s for a successful reach.

Single tungsten electrodes were used to record cortical activity from Monkeys P, A, and X and activity was recorded over the course of 127, 109, and 50 behavioral sessions, respectively. Floating micro-electrode arrays were used to record from Monkeys M, Pu, H, and C and neurons were included over the course of 3, 3, 1, and 1 behavioral sessions, respectively. Primary MC activity was recorded from Monkeys P, A, X, Pu, and M. Premotor cortex activity was also recorded from Monkeys P and A, which were pooled with the primary MC neurons. Primary somatosensory area 1 (areas 3 a and 1) and parietal area 5 were recorded from Monkey P. Primary somatosensory area 2 and parietal area 5 were recorded from Monkey A. Primary somatosensory area 2 was recorded from Monkeys H and C.

Spike timestamps were convolved with a gaussian kernel with a standard deviation of 30 ms. For displaying the single neuron responses only, timestamps were convolved with a half-gaussian kernel (SD 30 ms) that only estimated the instantaneous firing rate using spikes from the past. This prevented the appearance during the posture perturbation task that changes in firing rates preceded the onset of the load.

## Muscle recordings

Muscle activity was recorded percutaneously by inserting two single-stranded wires into the muscle belly (*Scott and Kalaska, 1997*). Stimulation was used to confirm the penetrated muscles. We recorded from the main extensor and flexor muscles of the shoulder and elbow including triceps (lateral and long), biceps (long and short), deltoids (anterior, medial, and posterior heads), brachioradialis, supraspinatus, and pectoralis major. From each monkey, we recorded a subset of these muscles that included a mixture of flexor and extensor muscles for both the shoulder and elbow joints.

## jPCA Analysis

We performed jPCA analysis on the neural network similar to *Churchland et al., 2012* using code available at https://churchland.zuckermaninstitute.columbia.edu/content/code. We constructed matrices X that contained the activities of all neurons in the network for every time point and condition (i.e., load combinations or reach directions). These matrices had NxCT dimensions, where N is the number of neurons in the network, C is the number of conditions, and T is the number of time points. The mean signal across conditions was subtracted at each time point and activity was soft normalized by the activity range plus a small constant (5e−4).

PCA was applied to X and the top-6 principle components were used to reduce X to $X_{Red}$ (6xCT dimensions). We numerically calculated the derivative of $X_{Red}$ yielding $\dot{X}_{Red}$ , and fit a linear dynamical model which found a relationship between $X_{Red}$ and $\dot{X}_{Red}$

$$\dot{X}_{Red} = MX_{Red} \tag{7}$$

where M is a 6×6 weight matrix. We assessed the model's fit by calculating the coefficient of determination ($R^2$).

With no constraint on M, any linear dynamical system could be captured by this equation including oscillators, point and line attractors, and so on. We compared how an unconstrained M performed with a fit where we constrained M to be skew-symmetric ($M_{Skew}$). This restricted the possible dynamical systems to systems with oscillatory dynamics. Skew-symmetric matrices have pairs of eigenvectors with eigenvalues that are complex conjugates of each other. These eigenvector pairs were found from $M_{Skew}$ and the corresponding activity generated two-dimensional jPCA planes. $M_{Skew}$ generates 3 jPCA planes and the planes were ranked by their eigenvalues (i.e., the speed of the rotational

dynamics) from highest to lowest. The amount of variance each plane captured of the original matrix X (VAF) was calculated and normalized by the total amount of variance in the original matrix X.

jPCA analysis was also applied to the kinematic feedback signals from the plant (normalization constant 0), the muscle activity produced by the network (0), the recorded neural activity (5sp/s), and the recorded EMG activity (0). Since there are fewer kinematic and muscle signals than neural signals, we only examined activity in the top-2 kinematic components, and the top-4 muscle components. For the posture task, jPCA analysis was applied for the first 300 ms after the load onset for the neural recordings. For the network, jPCA analysis was applied from 70 ms to 370 ms after the load onset to reflect the 50 ms delay in sensory feedback processing. Similar results were obtained using 0–300 ms epoch. For the reaching data, jPCA analysis was applied for the first 300 ms after the start of movement.

### Tensor maximum entropy

We tested our findings against the hypothesis that rotational dynamics are a byproduct of the tuning and smoothness properties of neurons. We employed TME to generate surrogate data sets (*Elsayed and Cunningham, 2017*) using code available at https://github.com/gamaleldin/TME, (*Kalidind, 2021* copy archived at swh:1:rev:ad1adf835e72dbba012406b5a3af30701adc8993). This method generates surrogate data sets that preserve the covariances across neurons, conditions, and time but not their interactions as required for rotational dynamics. Surrogate data sets were then sampled from this distribution and the jPCA analysis was applied to each data set (1000 iterations).

### Down-sampling neuron activity

For the muscle and kinematics, assessing whether the observed rotational dynamics were significant or not was complicated by the fact that there were fewer muscle and kinematics signals. Indeed, neural population dynamics deemed significant using TME were no longer significant after down-sampling the neural population to match the number of kinematic and muscle samples. Instead, we assessed whether the rotational dynamics in the muscle or kinematic signals were more dynamical than neural activity after correcting for the number of signals. We randomly sampled neurons from the neural population to match the number of muscles or kinematic signals and applied jPCA analysis to the resulting population activity. This was repeated 1000 times.

## Acknowledgements

The authors thank Kim Moore and Helen Bretzke for their laboratory and technical assistance. This work was supported by grants from the Canadian Institute of Health Research (PJT-159559). KPC was supported by an Ontario Graduate Scholarship. SHS was supported by a GSK chair in Neuroscience. HTK and EF are supported by the European Union's Horizon 2020 Framework Programme for Research and Innovation under the Specific Grant Agreement nos. 785907 (Human Brain Project SGA2) and 945539 (Human Brain Project SGA3).

## Additional information

### Competing interests

Stephen H Scott: Co-founder and CSO of Kinarm which commercializes the robotic technology used in the present study. The other authors declare that no competing interests exist.

### Funding

| Funder | Grant reference number | Author |
|---|---|---|
| Canadian Institutes of Health Research | PJT-159559 | Stephen H Scott |
| Horizon 2020 - Research and Innovation Framework Programme | 785907 (Human Brain Project SGA2) | Egidio Falotico |

| Funder | Grant reference number | Author |
|---|---|---|
| Horizon 2020 - Research and Innovation Framework Programme | 945539 (Human Brain Project SGA3). | Egidio Falotico |

The funders had no role in study design, data collection and interpretation, or the decision to submit the work for publication.

## Author contributions

Hari Teja Kalidindi, Kevin P Cross, Conceptualization, Formal analysis, Investigation, Methodology, Writing – original draft, Writing – review and editing; Timothy P Lillicrap, Conceptualization, Writing – original draft, Writing – review and editing; Mohsen Omrani, Conceptualization; Egidio Falotico, Funding acquisition, Supervision; Philip N Sabes, Writing – original draft, Writing – review and editing; Stephen H Scott, Conceptualization, Funding acquisition, Supervision, Writing – original draft, Writing – review and editing

## Author ORCIDs

Hari Teja Kalidindi (iD) http://orcid.org/0000-0003-2634-7953
Kevin P Cross (iD) http://orcid.org/0000-0001-9820-1043
Mohsen Omrani (iD) http://orcid.org/0000-0002-0461-1947
Philip N Sabes (iD) http://orcid.org/0000-0001-8397-6225
Stephen H Scott (iD) http://orcid.org/0000-0002-8821-1843

## Ethics

Studies were approved by the Queen's University Research Ethics Board and Animal Care Committee (#Scott-2010-035).

## Decision letter and Author response

Decision letter https://doi.org/10.7554/eLife.67256.sa1
Author response https://doi.org/10.7554/eLife.67256.sa2

---

## Additional files

### Supplementary files

• Transparent reporting form

### Data availability

The neural network code is publicly available at https://github.com/Hteja/CorticalDynamics (copy archived at https://archive.softwareheritage.org/swh:1:rev:d7f7bd80cb3be165f43ecc195308cf8f2f0b86e4). Data and analysis code is available at https://github.com/kevincross/CorticalDynamics Analysis (copy archived at https://archive.softwareheritage.org/swh:1:rev:d61decd3cd750147ef098de1041326fd2be07ab2).

The following previously published datasets were used:

| Author(s) | Year | Dataset title | Dataset URL | Database and Identifier |
|---|---|---|---|---|
| Chowdhury R, Glaser J, Miller L | 2020 | Data from: Area 2 of primary somatosensory cortex encodes kinematics of the whole arm | https://doi.org/10.5061/dryad.nk98sf7q7 | Dryad Digital Repository, 10.5061/dryad.nk98sf7q7 |

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
