## [Editor Report]

Motor cortical population activity during reaching exhibits rotational dynamics thought to arise from recurrent connections in cortical circuits. This innovative paper performs an important 'experiment' not currently possible in real biological networks: examine activity and task function before and after deletion of recurrent connections. Surprisingly, trained networks produced rotational dynamics even without internal recurrence, raising the possibility that sensory feedback is a key determinant of motor cortical dynamics. More broadly, this paper leverages the experimental tractability of artificial neural networks to test what conditions and architectures are necessary to produce brain-like signals.

---

## [Decision Letter]

**Decision letter after peer review:**

Thank you for submitting your article "Rotational dynamics in motor cortex are consistent with a feedback controller" for consideration by *eLife*. Your article has been reviewed by 3 peer reviewers, and the evaluation has been overseen by a Reviewing Editor and Richard Ivry as the Senior Editor. The reviewers have opted to remain anonymous.

Essential revisions:

1) The NO-REC Network

The point about non-recurrent (NO-REC) networks being able to produce rotational dynamics is critical in showing the key role of feedback in generation of those dynamics. Yet all three reviewers exhibited confusion and concern over *exactly* how the NO-REC was implemented.

In the Methods, the authors explain how they model a non-recurrent network as follows: "We also examined networks where we removed the recurrent connections from each layer by effectively setting W_hh_, W_oo_ to zero for the entire simulation and optimization (NO-REC networks)". However, if this is the only modification, it still leaves recurrent elements in the network. For example, if we set W_hh_ to zero, equation 2 will be: h_t+1_ = (1-a) * h_t_ + a * tanh(W_sh_ * s_t_ + b_h_) where a is a constant scalar (seems to be equal to 0.5). This is indeed still a recurrent neural network since h_t+1_ depends on h_t_. If their explanation in the Methods is accurate, then the current approach restricts the recurrent dynamics to be a specific linear dynamic (i.e. "h_t+1_ = (1-a) * h_t_ + …") but does not fully remove them. The second layer is also similar (equation 3) and will still have recurrent linear dynamics even if W_oo_ is set to 0. To be able to describe networks as non-recurrent, the first terms in equations 2 and 3 (that is (1-a)*h_t_ and (1-a)*o_t_) should also be set to 0. This is critical as an important argument in the paper is that non-recurrent networks can also produce rotational dynamics, so the networks supporting that argument must be fully non-recurrent, or the argument should be clarified and restricted to what was actually shown. Perhaps the authors have already done this but just didn't explain it in the Methods, in which case they should clarify the Methods. However, if the current Method description is accurate, they should rerun their NO-REC simulations by also setting the fixed linear recurrent components (that is (1-a)*h_t_ and (1-a)*o_t_) to zero as explained above to have a truly non-recurrent model.

2) Role of Sensory Feedback

A better discussion and handling of the distinction between recurrent feedback within M1 and recurrent feedback that traverses the whole system is necessary. All reviewers had issues with this, and all reviewers asked for *some* manipulation of the sensory feedback in the model to shore up the claim of its importance in driving rotations.

2.1 Reviewer 1 recommended deleting the feedback: To fully demonstrate the role of feedback, additional simulations are also needed where the sensory feedback is removed from the brain model. In other words, what would happen if recurrent and non-recurrent brain models are trained to perform the tasks but are not provided with the sensory feedback (only receive task goals)? One would expect the recurrent model to still be able to perform the task and autonomously produce similar rotational dynamics (as has been shown in prior work), but the non-recurrent model to fail in doing the task well and in showing rotational dynamics. I think adding such simulations without the feedback signals would really strengthen the paper and help its message.

2.2 Reviewer 2 additionally suggested tweaking the delays: asking what effect does the feedback delay have on the jPCA frequencies, and to test if increasing the delay leads to slower frequencies and decreasing the delay leads to faster frequencies.

2.3 Reviewer 3 asked if the sensory inputs were the positions and velocities of the two joints in Cartesian coordinates, would we observe similar rotational dynamics?

Full deletion of feedback (2.1) would probably make for the strongest response to this big point, and may also make testable predictions for cortical activity following de-afferentation. Yet we leave it at the author's discretion to choose which of these points to directly take on to bolster the investigation of how feedback affects the system – provided a clear argument in the response to reviewers is provided for why you chose to do what you did.

3. Network Performance Quality

A measure of how well each trained network is able to perform the task should be provided. For example, is the non-recurrent network able to perform the tasks as accurately as the recurrent models? The authors could use an appropriate measure, for example average displacement in the posture task and time-to-target in the center-out task, to objectively quantify task performance for each network. Another performance measure could be the first term of the loss in equation 5. Also, plots of example trials that show the task performance should be provided for the non-recurrent networks (for example by adding to Figure 8), similar to how they are shown for the recurrent models in Figures 2 and 6.

4. Role of Task Structure

An important observation is that rotational dynamics also exist in the sensory signals about the limb state. This may imply that the task structure that dictates the limb state and thus the associated sensory feedback may play an important role in the rotations without the recurrent connections. While the present study will be a valuable addition regardless of what the answer is, What is the role of the task structure in producing rotational dynamics? In both the posture task and the center-out task, the task instruction instructs subjects to return to the initial movement 'state' by the end of the trial: in the posture task the simulated arm needs to return to the original posture upon disturbance, and in the center out task the arm needs to start from zero velocity and settle at the target with zero velocity. Is this structure what's causing the rotational dynamics? This is an important question both for this paper and for the field and the authors have a great simulation setup to explore it. For example, what happens if the task instructions u* instruct the arm to follow a random trajectory continuously, instead of stopping at some targets? With a simulated tracking task like this, one could eliminate obvious cases of return-to-original-state from the task. Would the network still produce rotational dynamics? We do not expect the authors to collect experimental monkey data for such new tasks, rather to just change the task instructions in their numerical simulations to explore the dependence of observed rotational dynamics on the task structure. this will help the message of the paper and can be very useful for the field. If the authors choose not to address this point, there should be at least clear text in the discussion allowing for the possibility that their results might not generalize to other tasks, even though such a caveat might weaken the paper's argument.

*Reviewer #2 (Recommendations for the authors):*

I feel the study overloads the differences between recurrence and feedback. I am sure the authors do not mean to say that there is no recurrent processing in motor cortex! For instance, in the abstract the authors say:

Recent studies hypothesize that motor cortical (MC) dynamics are generated largely through its recurrent connections based on observations that MC activity exhibits rotational structure.

I think this claim is an overinterpretation of the Churchland 2012 and Sussillo 2015 articles from the Shenoy lab. The point is actually that when you apply jPCA to MC data, a surprising amount of it is rotational structure and one source of it can be local recurrence and that a lot of it can be described by autonomous dynamics. This past output could come through short low latency pathways within motor cortex. As these authors and Sussillo et al. 2015 argue, longer time constant pathways that involve the limb, and the joint muscles could be the cause of these rotations. Together, they all collaborate and result in the rotational structure in these brain areas.

I recognize the authors removed recurrent connections from the RNN and then show that a model without recurrence can replicate the rotational structure. I think this is somewhat imprecise because there is pseudo recurrence in the system except not in the traditional local synaptic sense but through a filtered version that comes through feedback from the output. In my mind what the authors have built is a sophisticated two-step albeit nonlinear autoregressive process which will show slow oscillations. To verify my idea, I developed a simple AR process simulation in MATLAB and observed that you can get oscillatory structure in the output even with an AR(2) and AR(4) process. The time scale of dt is ~10 ms and the time constant is 20 ms, and finally the feedback delay is ~50 ms which means this is essentially an AR(5) process which should lead to slow time scale changes in the network activity (please see attached code which simulates a 4th order AR process).

Let me try to make the point clearer. I feel the authors could really contribute by saying rotational structure in dynamical systems comes comes from past output affecting future output. In simple LDSes

dx/dt = Ax + U.

If A has complex eigen values, then you get rotational structure.

Now assume A = 0. Now somehow, if U contains a component which is some nonlinear form of x. Then you basically have a new equation which is

dx/dt = Af(x) + U'.

Now you have basically created another LDS which depends on the previous state. If A has complex eigen values you essentially have rotations again.

Related, what effect does the feedback delay have on the jPCA frequencies? It would be interesting to understand if increasing the delay leads to slower frequencies, and decreasing the delay leads to faster frequencies. This would further bolster the manuscript.

I think the discussion is very muddled and generic and is not as compelling as one would like it to be. In my mind, it seems weird to say recurrence has no role and also to think of MC as some sort of independent system divorced from the rest of the brain and the world. The way I formulate it is that the activity at the next time step in these motor related areas is a combination of local recurrent contribution, input from another cortical area, and feedback from the external world, which in itself is from another cortical area. All of these ultimately contribute to the rotational structure observed in motor cortex. This is the challenge for the field to understand all the constituent components. At best, this paper shows that just observing rotations means that any or all three of these components could be the cause. A better synthesis of their findings would improve the impact.

*Reviewer #3 (Recommendations for the authors):*

This work could be further strengthened with additional analyses of the trained RNNs that elucidate the conditions under which sensory inputs induce rotational dynamics in the MC.

1. The authors suggest that sensory inputs are likely the source of the rotational dynamics observed in the trained RNNs.

However, this is only weakly substantiated by linear fits of the activity in the top 2 jPCA planes to the sensory inputs.

Would any sensory inputs that are themselves rotational induce such phenomenon?

For example, if the sensory inputs were the positions and velocities of the two joints in Cartesian coordinates, would we observe similar rotational dynamics?

What if the feedback signals were not rotational at all (e.g., if we only provide the muscle activity as feedback signals)?

2. How does rotational dynamics in the trained RNNs depend on the length of the sensory delay? For example, would we expect a comparable delay to the onset of rotation dynamics in the RNN?

3. The authors showed that trained feedforward networks also exhibited rotational dynamics. Do trained recurrent connections converge to this solution? More generally, how strong is the internal activity of the trained RNNs as measured by for example the norm of the recurrent weights? Do the internal dynamics of the RNNs need to be weak for sensory inputs to induce rotational dynamics in the RNNs?

4. The authors seem to take for granted (perhaps rightfully so) that continuous feedback signals are necessary for performing the posture perturbation task. While this makes sense intuitively, it would help convince the readers to see evidence of this fact. More specifically, I assume RNNs trained with continuous feedback signals are likely able to generalize to perturbations that are not in the training set. On the other hand, RNNs trained without continuous feedback signals (but provided with the initial perturbed state of the arm) would not be able to generalize to novel perturbations.

5. The trained RNNs include an input layer and an output layer. Is this necessary? Would a single layer suffice?

---

## [Author Response]

Essential revisions:1) The NO-REC NetworkThe point about non-recurrent (NO-REC) networks being able to produce rotational dynamics is critical in showing the key role of feedback in generation of those dynamics. Yet all three reviewers exhibited confusion and concern over exactly how the NO-REC was implemented.In the Methods, the authors explain how they model a non-recurrent network as follows: "We also examined networks where we removed the recurrent connections from each layer by effectively setting Whh, Woo to zero for the entire simulation and optimization (NO-REC networks)". However, if this is the only modification, it still leaves recurrent elements in the network. For example, if we set Whh to zero, equation 2 will be: ht+1 = (1-a) * ht + a * tanh(Wsh * st + bh) where a is a constant scalar (seems to be equal to 0.5). This is indeed still a recurrent neural network since ht+1 depends on ht. If their explanation in the Methods is accurate, then the current approach restricts the recurrent dynamics to be a specific linear dynamic (i.e. "ht+1 = (1-a) * ht + …") but does not fully remove them. The second layer is also similar (equation 3) and will still have recurrent linear dynamics even if Woo is set to 0. To be able to describe networks as non-recurrent, the first terms in equations 2 and 3 (that is (1-a)*ht and (1-a)*ot) should also be set to 0. This is critical as an important argument in the paper is that non-recurrent networks can also produce rotational dynamics, so the networks supporting that argument must be fully non-recurrent, or the argument should be clarified and restricted to what was actually shown. Perhaps the authors have already done this but just didn't explain it in the Methods, in which case they should clarify the Methods. However, if the current Method description is accurate, they should rerun their NO-REC simulations by also setting the fixed linear recurrent components (that is (1-a)*ht and (1-a)*ot) to zero as explained above to have a truly non-recurrent model.

We thank the reviewer for raising this important concern. We have re-simulated the NO-REC network while removing the dynamics related to the leaky-integration component. This removal did not impact the network’s ability to perform the tasks and yielded virtually identical neural dynamics (see Figure 8). Throughout the Results we have updated the figures for the NO-REC network to the network without the leak-integration component.

2) Role of Sensory FeedbackA better discussion and handling of the distinction between recurrent feedback within M1 and recurrent feedback that traverses the whole system is necessary. All reviewers had issues with this, and all reviewers asked for some manipulation of the sensory feedback in the model to shore up the claim of its importance in driving rotations.2.1 Reviewer 1 recommended deleting the feedback: To fully demonstrate the role of feedback, additional simulations are also needed where the sensory feedback is removed from the brain model. In other words, what would happen if recurrent and non-recurrent brain models are trained to perform the tasks but are not provided with the sensory feedback (only receive task goals)? One would expect the recurrent model to still be able to perform the task and autonomously produce similar rotational dynamics (as has been shown in prior work), but the non-recurrent model to fail in doing the task well and in showing rotational dynamics. I think adding such simulations without the feedback signals would really strengthen the paper and help its message.

We apologize if the network architecture was not clear. In the case of the NO-REC network the only way they can generate the time-varying signals needed for the tasks is through sensory feedback. The network simply will not work without recurrent AND sensory feedback. For the posture task there are no additional inputs since it only receives sensory feedback. For the reaching task the task-goal input is static and the GO cue turns off on a timescale considerably shorter (~20ms) than the reach duration. Thus, the REC network would always perform better than the NO-REC network when sensory feedback was removed as the NO-REC network cannot generate any dynamics. We have now included in the Results the following statement. "Note, by removing the recurrent connections these networks can only generate time-varying outputs by exploiting the time-varying sensory inputs from the limb." (line 345-347).

We have also now included simulations to highlight how REC networks that receive sensory feedback are able to generalize better to scenarios with increased motor noise than REC networks where sensory feedback is either completely removed (reaching task) or only provided at the beginning of the trial (posture task) (Figure S8). Thus, sensory feedback makes REC networks more robust in less predictable scenarios.

2.2 Reviewer 2 additionally suggested tweaking the delays: asking what effect does the feedback delay have on the jPCA frequencies, and to test if increasing the delay leads to slower frequencies and decreasing the delay leads to faster frequencies.

We agree that this could be an interesting manipulation and have now included manipulations of the sensory feedback delays. We considered three separate delays, 0ms, 50ms and 100ms and found that there was a dependence on the rotational frequency of the top jPC plane with greater delays resulting in a general reduction in frequency (see now Supplementary Figure 10). There was less effect of delay on fit qualities to the constrained and unconstrained dynamical system. This has been added to the Results section (line 423-446).

2.3 Reviewer 3 asked if the sensory inputs were the positions and velocities of the two joints in Cartesian coordinates, would we observe similar rotational dynamics?

We simulated this scenario and found the answer to be rather complex and we have added these results to the supplementary material. The network's behavioural performance in the perturbation posture task is similar to the previous networks with joint-based feedback. However, the dynamics in the output layer are not the same with a clear reduction in how well the dynamics are described as rotational (Figure S11A-B). Oddly, rotational dynamics could still be observed in the input layer dynamics (data now shown) and the kinematic signals when they were converted to a cartesian reference frame (Figure S11D-E). Furthermore, rotational dynamics could emerge in the output layer if we used a different initialization method for the network weights. We initialized weights from a uniform distribution bound from ±1/N, where N is the number of units. In contrast, previous studies have initialized network weights using a Gaussian distribution with standard deviation equal to g/N where g is constant larger than 1. This alternative initialization scheme encourages strong intrinsic dynamics often needed for autonomous RNN models (Sussillo et al., 2015). We found networks initialized with this method and trained on the perturbation posture task exhibited stronger rotational dynamics with fits to the constrained and unconstrained dynamical systems of 0.5 and 0.88, respectively (Figure S11C-D). When examining the reaching task, we found similar results (Figure S11F-K). When initialized with a uniform distribution, fit quality for the constrained and unconstrained dynamical systems were 0.4 and 0.77, respectively (Figure S11F-G), which were smaller than for the joint-based feedback (Figure 7B, constrained R^2^=0.7, unconstrained R^2^=0.83). Qualitatively, the dynamics were different when the network was initialized with a Gaussian distribution (Figure S11H), however fit qualities were comparable between the two initialization methods (Figure S11 I). There was also a noticeable reduction in the fit quality for the kinematic signals particularly for the constrained dynamical system (Figure S11K, constrained R^2^=0.36, unconstrained R^2^=0.77). These findings have been added to the Results.Full deletion of feedback (2.1) would probably make for the strongest response to this big point, and may also make testable predictions for cortical activity following de-afferentation. Yet we leave it at the author's discretion to choose which of these points to directly take on to bolster the investigation of how feedback affects the system – provided a clear argument in the response to reviewers is provided for why you chose to do what you did.

3. Network Performance QualityA measure of how well each trained network is able to perform the task should be provided. For example, is the non-recurrent network able to perform the tasks as accurately as the recurrent models? The authors could use an appropriate measure, for example average displacement in the posture task and time-to-target in the center-out task, to objectively quantify task performance for each network. Another performance measure could be the first term of the loss in equation 5. Also, plots of example trials that show the task performance should be provided for the non-recurrent networks (for example by adding to Figure 8), similar to how they are shown for the recurrent models in Figures 2 and 6.

We have now presented and quantified the NO-REC network behavioural performance. Kinematics for the NO-REC network are shown in Figure S7A-C and E-G which are comparable to the REC network. Furthermore, quantifying the maximum displacement during the posture task yielded no obvious differences between the NO-REC and REC networks (Figure S7D). For the reaching task, the time-to-target was noticeably more variable and tended to be slower for the NO-REC network (Figure S7H). These observations have been added to the Results.

4. Role of Task StructureAn important observation is that rotational dynamics also exist in the sensory signals about the limb state. This may imply that the task structure that dictates the limb state and thus the associated sensory feedback may play an important role in the rotations without the recurrent connections. While the present study will be a valuable addition regardless of what the answer is, What is the role of the task structure in producing rotational dynamics? In both the posture task and the center-out task, the task instruction instructs subjects to return to the initial movement 'state' by the end of the trial: in the posture task the simulated arm needs to return to the original posture upon disturbance, and in the center out task the arm needs to start from zero velocity and settle at the target with zero velocity. Is this structure what's causing the rotational dynamics? This is an important question both for this paper and for the field and the authors have a great simulation setup to explore it. For example, what happens if the task instructions u* instruct the arm to follow a random trajectory continuously, instead of stopping at some targets? With a simulated tracking task like this, one could eliminate obvious cases of return-to-original-state from the task. Would the network still produce rotational dynamics? We do not expect the authors to collect experimental monkey data for such new tasks, rather to just change the task instructions in their numerical simulations to explore the dependence of observed rotational dynamics on the task structure. this will help the message of the paper and can be very useful for the field. If the authors choose not to address this point, there should be at least clear text in the discussion allowing for the possibility that their results might not generalize to other tasks, even though such a caveat might weaken the paper's argument.

We agree that a tracking task would be an interesting manipulation and have simulated this with the REC and NO-REC networks (Figure 9). Here, we trained up the network to reach from the starting position and track a target moving radially at a constant velocity for the rest of the trial (1.2seconds). Thus, the network has to move the limb at a constant velocity. We found there was a consistent reduction in how well the network’s dynamics (constrained R^2^=0.13, unconstrained R^2^=0.3) were described as rotational when compared to the previous reaching task (Figure 7, constrained R^2^=0.7, unconstrained R^2^=0.83). Also, note that this reduction in rotational dynamics remained even when we initialized the network weights using a Gaussian distribution (see Essential revision 2.3). These simulations have been added to the Results section.

Reviewer #2 (Recommendations for the authors):I feel the study overloads the differences between recurrence and feedback. I am sure the authors do not mean to say that there is no recurrent processing in motor cortex! For instance, in the abstract the authors say:Recent studies hypothesize that motor cortical (MC) dynamics are generated largely through its recurrent connections based on observations that MC activity exhibits rotational structure.I think this claim is an overinterpretation of the Churchland 2012 and Sussillo 2015 articles from the Shenoy lab. The point is actually that when you apply jPCA to MC data, a surprising amount of it is rotational structure and one source of it can be local recurrence and that a lot of it can be described by autonomous dynamics. This past output could come through short low latency pathways within motor cortex. As these authors and Sussillo et al. 2015 argue, longer time constant pathways that involve the limb, and the joint muscles could be the cause of these rotations. Together, they all collaborate and result in the rotational structure in these brain areas.

We agree with the reviewer that we were not clear enough throughout the article about what we meant with recurrent connections. The main issue we raise in this paper is that many groups assume that these rotational dynamics emerge from intrinsic recurrent connections between neurons in MC with little influence from recurrent sensory feedback. The abstract now reads "Recent studies have identified rotational dynamics in the activity patterns of motor cortex (MC) which many assume arise from the intrinsic connections of MC". Throughout the paper we have also made edits to clarify that we are focused on the intrinsic recurrent connections.

We also agree that the motor system must include intrinsic processing generated from intrinsic connections. These intrinsic connections may include local connections in motor cortex, but also involve contributions from areas reciprocally connected with motor cortex including the cerebellum and somatosensory cortex. We have included a paragraph in the Discussion about this which reads (lines 572-583):

"However, it is important to stress that intrinsic processing during motor actions is also required for optimal feedback control. Optimal feedback controllers relay an efference copy of the motor commands to circuits involved with state estimation that estimate the future state of the limb using an internal model. This estimate is combined with sensory feedback and sent back to circuits involved with implementing the control policy (i.e. MC), and thus, creating a recurrent loop. Optimal feedback controllers also generate time-varying feedback gains that map the limb estimate to the desired motor commands (Dimitriou et al., 2013; Liu and Todorov, 2007). These feedback gains must be generated independent of the current limb estimate, and thus, requires dynamics that are generated intrinsically. Although in theory this intrinsic processing could be done by local synaptic connections in MC, it is likely that intrinsic processing inside feedback loops between cortical and subcortical areas such as the cortico-cerebellar-cortical loops play an important role"

I recognize the authors removed recurrent connections from the RNN and then show that a model without recurrence can replicate the rotational structure. I think this is somewhat imprecise because there is pseudo recurrence in the system except not in the traditional local synaptic sense but through a filtered version that comes through feedback from the output. In my mind what the authors have built is a sophisticated two-step albeit nonlinear autoregressive process which will show slow oscillations. To verify my idea, I developed a simple AR process simulation in MATLAB and observed that you can get oscillatory structure in the output even with an AR(2) and AR(4) process. The time scale of dt is ~10 ms and the time constant is 20 ms, and finally the feedback delay is ~50 ms which means this is essentially an AR(5) process which should lead to slow time scale changes in the network activity (please see attached code which simulates a 4th order AR process).Let me try to make the point clearer. I feel the authors could really contribute by saying rotational structure in dynamical systems comes comes from past output affecting future output. In simple LDSesdx/dt = Ax + U.If A has complex eigen values, then you get rotational structure.Now assume A = 0. Now somehow, if U contains a component which is some nonlinear form of x. Then you basically have a new equation which isdx/dt = Af(x) + U'.Now you have basically created another LDS which depends on the previous state. If A has complex eigen values you essentially have rotations again.Related, what effect does the feedback delay have on the jPCA frequencies? It would be interesting to understand if increasing the delay leads to slower frequencies, and decreasing the delay leads to faster frequencies. This would further bolster the manuscript.

See Essential Revision 2

I think the discussion is very muddled and generic and is not as compelling as one would like it to be. In my mind, it seems weird to say recurrence has no role and also to think of MC as some sort of independent system divorced from the rest of the brain and the world. The way I formulate it is that the activity at the next time step in these motor related areas is a combination of local recurrent contribution, input from another cortical area, and feedback from the external world, which in itself is from another cortical area. All of these ultimately contribute to the rotational structure observed in motor cortex. This is the challenge for the field to understand all the constituent components. At best, this paper shows that just observing rotations means that any or all three of these components could be the cause. A better synthesis of their findings would improve the impact.

We agree with the reviewer that thinking about the evolution of motor cortical dynamics as a combination of intrinsic processing and inputs from the external world is the correct approach for the field. We have articulated this point in the Discussion which now reads (lines 584-587):

"Thus, from the OFC perspective the rotational dynamics in MC are generated by a number of factors including local intrinsic connections, inputs from other brain areas and inputs from sensory feedback. Teasing apart how these factors combine and give rise to the observable dynamics remains an important open question. "

We have also now also included a paragraph highlighting that OFC predicts that the motor system must exploit intrinsic processing for good control (see above). We have also made several other changes to the Discussion to try and better highlight our findings.

Reviewer #3 (Recommendations for the authors):This work could be further strengthened with additional analyses of the trained RNNs that elucidate the conditions under which sensory inputs induce rotational dynamics in the MC.1. The authors suggest that sensory inputs are likely the source of the rotational dynamics observed in the trained RNNs.However, this is only weakly substantiated by linear fits of the activity in the top 2 jPCA planes to the sensory inputs.Would any sensory inputs that are themselves rotational induce such phenomenon?For example, if the sensory inputs were the positions and velocities of the two joints in Cartesian coordinates, would we observe similar rotational dynamics?What if the feedback signals were not rotational at all (e.g., if we only provide the muscle activity as feedback signals)?

See Essential Revision 2

2. How does rotational dynamics in the trained RNNs depend on the length of the sensory delay? For example, would we expect a comparable delay to the onset of rotation dynamics in the RNN?

See Essential Revision 2

3. The authors showed that trained feedforward networks also exhibited rotational dynamics. Do trained recurrent connections converge to this solution? More generally, how strong is the internal activity of the trained RNNs as measured by for example the norm of the recurrent weights? Do the internal dynamics of the RNNs need to be weak for sensory inputs to induce rotational dynamics in the RNNs?

We agree examining the weights of the REC network could be useful. The following analysis has been added to the Results (lines 387-406):

"For each neural unit in the input layer of the REC network, we examined their synaptic weights for the sensory feedback connections and intrinsic recurrent connections from neurons in the layer (Figure S9A). We computed the ratio between the norm of the local recurrent connections and the sensory feedback connections for each individual neuron (Figure S9B, C). The resulting distribution was slightly larger than one indicating weights were larger for the recurrent connections than the feedback connections. However, examining the relative synaptic weights can be misleading as the total contribution of an input to a neuron's response is the product of the weight with the activity of the input signal (Figure S9A, E[s] and E[r]). Thus a more appropriate comparison is to compare the currents (W*E[∙]) generated from intrinsic and sensory sources. Figure S9E and F show the ratio of the intrinsic currents with the sensory currents across neurons. The distributions are centered near 0.5 indicating that the sensory contribution is ~2x larger than the recurrent contribution across neurons. Thus, sensory inputs had a substantial impact in generating the dynamics in the REC networks. However, we caution interpreting these results in the context of a biological system as many factors not modeled will likely contribute to the relative weighting of intrinsic dynamics and sensory feedback."

4. The authors seem to take for granted (perhaps rightfully so) that continuous feedback signals are necessary for performing the posture perturbation task. While this makes sense intuitively, it would help convince the readers to see evidence of this fact. More specifically, I assume RNNs trained with continuous feedback signals are likely able to generalize to perturbations that are not in the training set. On the other hand, RNNs trained without continuous feedback signals (but provided with the initial perturbed state of the arm) would not be able to generalize to novel perturbations.

We agree with the reviewer. In the Results we now compare networks that received sensory feedback continuously for the entire trial with networks that either received no sensory feedback or were provided sensory feedback at the beginning of the trial for the first 200ms. We challenged these networks after training by increasing the amount of motor noise and examined the resulting impact on behavioural performance. Note, the networks were not trained with motor noise present, motor noise was only added after training was complete. Consistently, the network that received continuous sensory feedback resulted in better control and lower endpoint dispersion for both the perturbation posture task and the reaching task. These findings have been added to the Results section (lines 360-384) and as Supplementary Figure 8.

5. The trained RNNs include an input layer and an output layer. Is this necessary? Would a single layer suffice?

We have observed similar rotational dynamics when we used only a single layer network (see Author response image 1). The main reason we used a two-layer configuration for the network is that it tends to converge faster to a solution and is consistent with our previous work involving these neural networks (Lillicrap and Scott 2013). In the Methods we now include the following statement (lines 668-670):

"Note, although we used a two-layer network, similar results were found using a one-layer network, however the training time needed to converge to a solution tended to be longer in the one-layer network."

**Author response image 1. sa2fig1:**